# ChartMoE: Mixture of Diversely Aligned Expert Connector for Chart Understanding

**Zhengzhuo Xu**[12*] **Bowen Qu**[13*] **Yiyan Qi**[1*] **Sinan Du**[2] **Chengjin Xu**[1] **Chun Yuan**[2†] **Jian Guo**[14†]

[1]International Digital Economy Academy  [2]Tsinghua University  [3]Peking University
[4]Hong Kong University of Science and Technology (Guangzhou)
https://github.com/IDEA-FinAI/ChartMoE

## Abstract

Automatic chart understanding is crucial for content comprehension and document parsing. Multimodal Large Language Models (MLLMs) have demonstrated remarkable capabilities in chart understanding through domain-specific alignment and fine-tuning. However, current MLLMs still struggle to provide faithful data and reliable analysis only based on charts. To address it, we propose ChartMoE, which employs the Mixture of Expert (MoE) architecture to replace the traditional linear projector to bridge the modality gap. Specifically, we train several linear connectors through distinct alignment tasks, which are utilized as the foundational initialization parameters for different experts. Additionally, we introduce ChartMoE-Align, a dataset with nearly 1 million chart-table-JSON-code quadruples to conduct three alignment tasks (chart-table/JSON/code). Combined with the vanilla connector, we initialize different experts diversely and adopt high-quality knowledge learning to further refine the MoE connector and LLM parameters. Extensive experiments demonstrate the effectiveness of the MoE connector and our initialization strategy, e.g., ChartMoE improves the accuracy of the previous state-of-the-art from 80.48% to 84.64% on the ChartQA benchmark.

## 1 Introduction

Charts serve as a fundamental tool for data visualization, with automated chart interpretation gaining prominence in domains such as text analysis (Hoque et al., 2017), scientific research (Hsu et al., 2021), and policy-making (Wu et al., 2024). Chart understanding is a complex task that demands the identification of visual cues, the comprehension of intricate interactions, and the precise inference of values informed by prior knowledge (Huang et al., 2024). Previous work (Liu et al., 2023b;a) typically pre-trained on domain-specific charts, which are constrained by limited resources and narrow task focus. In contrast, Multi-modal Large Language Models (MLLMs) (Li et al., 2023; Liu et al., 2023d; Bai et al., 2023a; Ye et al., 2023b; Chen et al., 2023; OpenAI, 2023) exhibit substantial potential in image comprehension and instruction following. The community has achieved advanced progress by creating chart understanding datasets (Liu et al., 2023c; Han et al., 2023; Masry et al., 2024b; Xu et al., 2023) and applying supervised fine-tuning based on well-performed MLLMs (Meng et al., 2024; Yan et al., 2024). With the exponential growth of chart data, automated chart interpretation via MLLMs is emerging as a promising avenue.

Recent studies advocate for chart alignment as a foundational step for LLaVA-like MLLMs (Liu et al., 2023d; Zhang et al., 2023; Xue et al., 2024), which bridge the visual encoder and LLM through MLP connector. They usually utilize chart-to-table alignment task to train the connector effectively (Meng et al., 2024; Yan et al., 2024; Hu et al., 2024). However, tables only provide basic information, such as numerical values and titles, which fail to capture the full range of chart elements. Despite some efforts to align with more informative text (Yan et al., 2024), the heavy alignment tasks may lead to the erosion of the connector's general capabilities, e.g., instruction following and visual counting, which are derived from the pre-training on large-scale visual-language data. To mitigate knowledge forgetting, one intuitive approach is to further tune with its original data, which results in redundant training and computational burden.

In this paper, we try to address these challenges via Mixture of Experts (MoE) architecture (Zoph et al., 2022). MoE enhances model capacity by activating a subset of experts through a router. Since

---

*Equal contributions. †Corresponding authors.

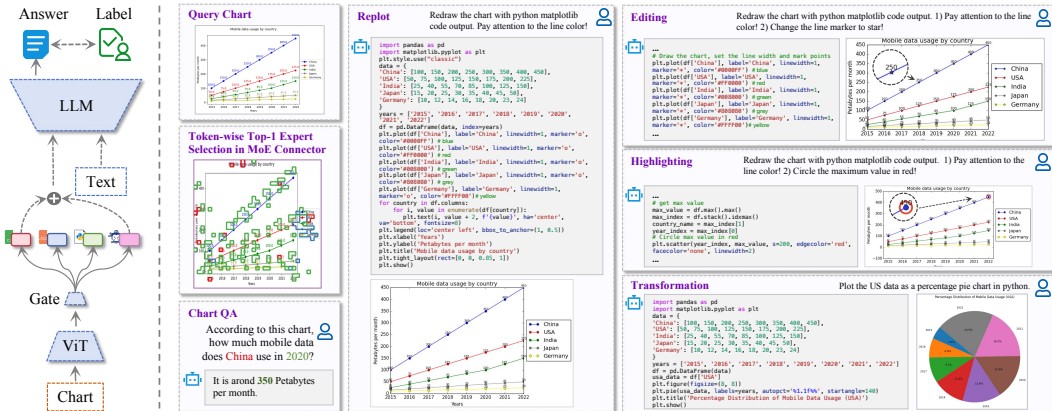

Figure 1: Overview and capabilities of ChartMoE: We introduce a MoE architecture connector and provide visualizations of the top-1 expert selection (refer to Fig. 6 and Appendix B for details). ChartMoE can extract highly precise values and provide flexible chart editing through code-based interactions.

the alignment tasks work on the connector, we replace only the MLP projector with MoE while keeping the vision encoder and LLM frozen. Our insight lies in the ***expert initialization manner***. Random initialization can lead to training instability and convergence at sub-optimal points (Fig. 4). Recent co-upcycling initialization (Komatsuzaki et al., 2023) addresses this issue by duplicating the vanilla connector parameters across all experts. However, it fails to avoid the dilemma of expert homogenization, where the experts end up with similar functionalities.

In contrast, we attempt to inject distinct prior knowledge into each expert first to tackle these challenges. Unlike natural images, *charts can be represented in various text formats, e.g., tables, attribute JSON, and rendering code*. As shown in Fig. 1& 2, in addition to chart-table, we introduce chart-JSON alignment to capture detailed elements like color or topological relationships and chart-code alignment to incorporate rendering details such as numerical values, color hex codes, and visual elements interactions (refer to Appendix C). We independently conduct various alignment tasks to capture more diverse chart features and thus obtain three distinct initialization approaches. We also retain the vanilla connector to preserve the capabilities of the MLLM on general tasks effectively.

Building upon the proposed four initialization manners, we introduce ChartMoE, an SFT-based MLLM with MoE connector for chart comprehension and reasoning. Our purpose in applying MoE connector is not to increase model capacity, but rather to improve chart comprehension and diverse representation through alignment tasks, while maintaining performance on other general tasks. Hence, we preserve the original connector parameters as one expert initialization manner. The MoE connector is extremely lightweight, so it adds negligible computational cost during both training and inference. Interestingly, we observe that experts in ChartMoE exhibit distinct visual token preferences, e.g., the vanilla expert favors background tokens while other experts focus more on tokens with legends or numbers (Fig. 6 and Appendix B). Considering that the distribution of visual tokens is naturally imbalanced in chart scenarios, we remove the expert-balanced loss in MoE and obtain further performance gain. Due to the scarcity of rich structural text for chart alignment, we design a pipeline (Fig. 3) to generate nearly 1 million quadruplets chart-table-JSON-code to build the ChartMoE-Align dataset for alignment. We train ChartMoE in 3 stages. First, we initialize experts via the proposed four manners. Then, we conduct high-quality knowledge learning using the MMC instruction (Liu et al., 2023c) to train the routing network, expert connectors, and LoRA (Hu et al., 2022) modules. Finally, we employ annealing training on ChartQA (Masry et al., 2022) and ChartGemma (Masry et al., 2024b). ChartMoE achieves state-of-the-art (SOTA) performance and provides more precise numbers and comprehensive attributes of charts (Appendix E). Refer to Appendix D.1 for detailed comparisons with other MoE works w.r.t. *motivation*, *initialization*, and *complexity*. In summary, our contributions are:

a) We present ChartMoE for faithful and reasonable chart understanding, with the connector based on Mixture of Expert architecture, to bridge the chart and LLM branches. All experts are initialized based on various alignment training tasks to avoid expert homogenization.

b) We introduce ChartMoE-Align, a large-scale dataset with nearly 1 million meticulous chart-table-JSON-code quadruplets for chart alignment pre-training.

c) We propose to train ChartMoE with a three-stage training paradigm, including connector alignment pre-training, high-quality knowledge learning, and annealing chart tuning.

d) Extensive quantitative and qualitative studies demonstrate that ChartMoE significantly outperforms previous state-of-the-art across several benchmarks by a large margin.

## 2 RELATED WORK

**Multimodal large language models** leverages a connector to bridge the gap between large language models (Touvron et al., 2023; Radford et al., 2018; Brown et al., 2020; Zhang et al., 2022; Zheng et al., 2023) and vision encoders (Radford et al., 2021; Oquab et al., 2023) to enable enriched capabilities of comprehension and instruction following. Approaches such as BLIP2 (Li et al., 2023), Flamingo (Alayrac et al., 2022), mPLUG-Owl (Ye et al., 2023b), and Qwen-VL (Bai et al., 2023b) utilize QFormers or Resamplers to align modalities on extensive datasets of image-text pairs. LLaVA (Liu et al., 2023d; 2024b) is the pioneering work to extend the instruction tuning paradigm to visual tasks with text-only GPT-4 (OpenAI, 2023), achieving tremendous performance using a simple MLP without compromising visual information to refine the multimodal alignment. Some works (Lin et al., 2023; Tong et al., 2024b;a) explore the combination of various vision encoders, complementarily enhancing visual representations to bolster the fine-grained visual perception of MLLMs. Despite efforts in structural design, training strategies and data quality remain crucial in the advancement of MLLMs.

**Chart Reasoning** refers to chart analysis, summarization, and etc. Existing methods can be categorized as 1) *Two-stage methods* use specialized extraction modules to generate intermediary representations of chart information, like tables, which are provided as textual prompts for LLMs. Pix2Struct (Lee et al., 2023) aligns markdown data with charts. MatCha (Liu et al., 2023b) aligns various data formats (e.g., tables and code) with charts on several downstream tasks. DePlot (Liu et al., 2023a) fine-tunes Pix2Struct for table extraction and uses LLMs to process queries based on the extracted data. ChartVLM (Xia et al., 2024) employs a discriminator to ascertain the necessity of intervention by LLMs for a given query. 2) *End-to-end methods* strive to tackle chart reasoning challenges with a unified model. ChartLlama (Han et al., 2023) incorporates diverse charts and downstream tasks based on LLaVA (Liu et al., 2023d). ChartPaLI (Carbune et al., 2024), ChartAst (Meng et al., 2024), and MMC (Liu et al., 2023c) conduct alignment on table-chart pairs. UReader (Ye et al., 2023a) aligns all data with markdown, while mPLUG-Owl2 (Ye et al., 2023c) achieves superior performance with high-resolution inputs. ChartThinker (Liu et al., 2024c) and DOMINO (Wang et al., 2023) propose the CoT (Wei et al., 2022) for chart reasoning. LaMenDa (Zhuowan et al., 2024) trains MLLMs via step-by-step reasoning QA. ChartReformer (Yan et al., 2024) introduces chart-JSON alignment, while OneChart (Chen et al., 2024) aligns charts with Python dictionaries. MiniGPT-v2 (Chen et al., 2023), Doc-Owl (Hu et al., 2024), and TinyChart (Zhang et al., 2024) tackle the reasoning efficiency for high-resolution charts by merging tokens.

## 3 CHARTMOE

### 3.1 ARCHITECTURE

The ChartMoE is based on InternlmXC-v2 (Dong et al., 2024) due to the concise LLaVA-like architecture (Liu et al., 2023d) and performance on par with GPT-4 on text-image comprehension. The base model includes a vision encoder and a LLM connected by a two-layer MLP. ChartMoE replaces the MLP with a MoE architecture as the connector to leverage diverse prior knowledge.

**Vision Encoder.** We utilize CLIP ViT-Large (Radford et al., 2021) as the vision encoder, leveraging its rich prior knowledge gained from training on millions of image-text pairs. Considering the impact of chart resolution on performance, we set the input resolution to $490 \times 490$ to strike a balance between efficiency and performance. Formally, the visual encoder $\mathcal{M}^V(\cdot)$ will project the chart $\mathcal{I}$ into $N$ tokens $V := \{v_1, v_2, \ldots, v_N\}$, where $N = 1225$ in the ChartMoE.

**Mixture-of-Experts Connector.** As illustrated in Fig. 2c, the MoE architecture employs a parallel multi-expert collaboration approach. This architecture comprises $L$ experts $\mathcal{M}^E(\cdot)$, each designed with the same linear layer as the baseline. For a visual token $v_i$ given by $\mathcal{M}^V$, the gating network $\mathcal{M}^G(\cdot)$ will calculate the routing weight $g_j(v_i)$ of each expert $\mathcal{M}^E_j(\cdot)$ and select top-$K$ to activate.

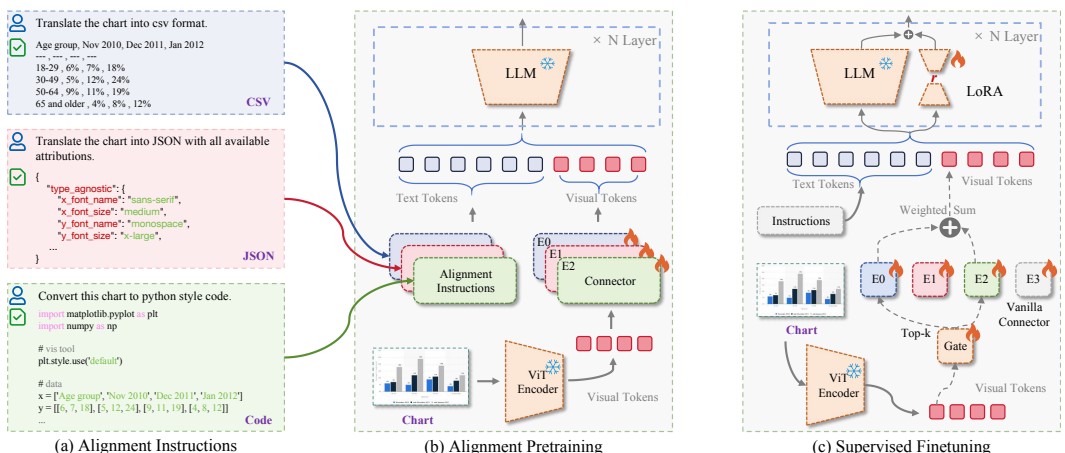

Figure 2: Overview of proposed ChartMoE. (a) Examples of alignment instructions. (b) We conduct three different alignment tasks in parallel. (c) We initialize MoE connectors in four different manners and train the gate network, experts, and LoRA during the supervised fine-tuning stage.

Finally, the tokens processed by each expert $\mathcal{M}_j^E$ will be averaged according to the weight $g_j(v_i)$ given by $\mathcal{M}^G$ to get the token $\hat{v}_i$ for the LLM branch $\mathcal{M}^L$.

**Large Language Model.** Following the baseline, we employ the *InternLM2-7B-ChatSFT* variant as the LLM $\mathcal{M}^L$, implemented as a transformer decoder with a causal attention mask. We concate the visual tokens $\hat{V} := \{\hat{v}_1, \hat{v}_2, \ldots, \hat{v}_N\}$ given by MoE connector with the $M$ input text $\mathcal{T}$ tokens $T := \{t_1, t_2, \ldots, t_M\}$ to form the input token sequence for the LLM $\mathcal{M}^L$. Formally, given the chart $\mathcal{I}$ and instruction $\mathcal{T}$, the output $\mathcal{O}$ of proposed ChartMoE can be formulated as:

$$\{v_1, v_2, \ldots, v_N\} = \mathcal{M}^V(\mathcal{I}), \tag{1}$$

$$\hat{v}_i = \sum_j^L g_j(v_i)\mathcal{M}_j^E(v_i), \quad g(v_i) = \text{Top}(\sigma(\mathcal{M}^G(v_i)); K), \tag{2}$$

$$\mathcal{O} = \mathcal{M}^L(\{\hat{v}_1, \hat{v}_2, \ldots, \hat{v}_N; t_1, t_2, \ldots, t_M\}), \tag{3}$$

where $\sigma$ indicates *softmax* and the $\text{Top}(\cdot; K)$ will reset the non-Top K routing weight to 0.

**Initialization of Expert.** Previous approaches initialize expert parameters via *1) Random initialization*, which may lead to convergence difficulties during supervised fine-tuning, and *2) Co-upcycling initialization* (Komatsuzaki et al., 2023), i.e., copy baseline connector parameters to each expert, which may lead to homogenization of experts. ChartMoE proposes initializing experts' parameters through distinct alignment tasks. We eliminate the load-balancing loss typically used in standard MoE architectures to equalize expert activation frequencies, as our initialization approach allows each expert to specialize in its preferred visual tokens, which inherently exhibit biased distributions.

## 3.2 ALIGNMENT PRE-TRAINING.

The key insight of ChartMoE is the experts' initialization parameters from the different alignment pre-training (Fig. 2a). Specifically, as illustrated in Fig. 2b, we align expert connectors using three distinct alignment tasks, where only the connector parameters will be updated. We visualize the *visual token preferences* of each expert for both chart (Fig. 6 & 12) and non-chart (Fig. 11) images.

**Alignment with Table.** Charts convey key information that can be more precisely expressed in tabular form, and LLMs are particularly adept at processing such structured data. Hence, we introduce a chart-table alignment task, aiming to translate chart content into tabular format. The connector is trained to convert chart information into corresponding CSV tables, thereby improving model performance in numerical data extraction and chart interpretation.

**Alignment with JSON.** Although tables capture the numerical information from charts, they miss semantic details such as colors, shapes, and fonts. To fill this gap, we propose a chart-JSON alignment task, which represents chart attributes in JSON format. This task requires the connector to

Figure 3: Overview of ChartMoE-Align data generation pipeline. The charts are plotted by Python *matplotlib*.

focus not only on the numerical data but also on visual and semantic properties. Accurately extracting chart attributes is essential for tasks like chart redrawing and editing.

**Alignment with Code.** To fully align with charts, we further introduce a chart-code alignment task. Since the underlying drawing code fully defines a chart, this approach enables the connector to convert the chart's visual tokens into representations in the LLM domain. Notably, we provide the drawing code explicitly, including precise numerical values and rendering attributes, e.g., numbers represented in Python lists and colors in hexadecimal code. Refer to Fig. 13 for more detailed cases. The code enables the model to perform in-depth summarization, analysis, and editing of charts. This expert is significantly more sensitive to the trends and key elements in the charts.

**ChartMoE-Align generation pipeline.** As Fig. 3 illustrates, 1) We filter charts with meta CSV from existing datasets (Masry et al., 2022; Methani et al., 2020) and data generated by LLMs (Chen et al., 2024). 2) We use a fine-tuned Deplot (Liu et al., 2023a) to inverse the plotting attributes following the templates provided by ChartReformer and randomly sample missing attributes from the predefined set. 3) We create code templates for different types of charts

Table 1: Datasets used for training ChartMoE. We conduct alignment pre-training with synthetic data and supervised tuning with high-quality, real-world data. Only ChartQA is used in the ablation due to GPU constraints.

| Source | Data Type | Task Type | Samples |
|---|---|---|---|
| *Alignment Training* | | | |
| ChartQA | synthetic | chart to table | 18.3K |
| | | chart to JSON | 18.3K |
| | | chart to code | 18.3K |
| PlotQA | synthetic | chart to table | 157K |
| | | chart to JSON | 157K |
| | | chart to code | 157K |
| ChartY | synthetic | chart to table | 763.6K |
| | | chart to JSON | 763.6K |
| | | chart to code | 763.6K |
| Total | | | 2.8M |
| Usage: Table = 500K JSON = 200K Code = 100K | | | 800K |
| *High-Quality Knowledge Learning* | | | |
| MMC | synthetic & real world | QA & reasoning & summariztion | 410K |
| *Chart Specific Annealing Tuning* | | | |
| ChartQA | real world | QA | 28.3K×2 |
| ChartGemma | real world | QA & PoT & reasoning & summariztion | 163.2K |
| Total | | | 220.8K |

and generate plotting code based on the meta CSV and extracted JSON attributes. *Note that all values and attributes in the code are explicitly represented.* 4) We retain the (table, JSON, code, chart) quadruples that pass compilation. Tab. 1 shows the data sources & size and refer to Appendix C for details.

## 3.3 SUPERVISED FINE-TUNING.

We initialize ChartMoE using the structure shown in Fig. 2c after aligning the connectors across 3 distinct tasks separately. We also retain the vanilla connector to maintain the baseline's excellent dialogue capabilities, which aligns with the principle of residual optimization (He et al., 2016). We train the MoE connector and LLM during this stage with LoRA (Hu et al., 2022), as shown in Fig. 2c. Considering the training principles proposed in LLaVA-NeXT (Liu et al., 2024a), this stage is divided into high-quality knowledge learning and chart-specific annealing training.

**High-Quality Knowledge Learning.** We adopt MMC (Liu et al., 2023c) to enhance the ChartMoE's knowledge. MMC includes a variety of chart types and tasks such as chart-related question answering, translation, extraction, reasoning, and analysis. Considering data quality, we only utilize the MMC-Instruction subset, which has been manually verified. Notice that the quality of instruction data is more important than quantity in this stage.

**Chart Specific Annealing Tuning.** Following Llama-v3.1 (Team et al., 2024b), we perform annealing tuning before evaluating mainstream benchmarks. We increase the learning rate and conduct instruction tuning using the training sets of ChartQA and ChartGemma to adjust the query styles and answer formats of these benchmarks.

**Program of Thought (PoT) Inference.** We require the model to generate the variables and operation code rather than producing direct answers. This inference pipeline addresses the mathematical capabilities by employing *Python* to handle the logical computations, which is the shortcoming of

Table 2: The relaxed accuracy (%) performance on ***ChartQA***. Ada.: Adaptive input resolution. *: Multi-scale image feature, 448×448 in DocOwl. †: Employing token merging to reduce computational overhead.

| Models | Para. | Resolution | Relax Acc. @0.05 | | | Relax Acc. @0.10 | | | Relax Acc. @0.20 | | |
|---|---|---|---|---|---|---|---|---|---|---|---|
| | | | Human | Aug. | Avg. | Human | Aug. | Avg. | Human | Aug. | Avg. |
| General MLLMs | | | | | | | | | | | |
| LLaVA-v1.5 | 13B | @336 | 25.36 | 18.56 | 21.96 | 28.56 | 23.52 | 26.04 | 32.56 | 30.72 | 31.64 |
| Qwen-VL | 9.6B | @448 | 40.48 | 79.76 | 60.12 | 43.20 | 82.56 | 62.88 | 47.52 | 85.76 | 66.64 |
| DocOwl-v1.5 | 8B | @448* | 47.44 | 91.52 | 72.00 | 51.92 | 92.08 | 72.00 | 56.72 | 93.12 | 74.92 |
| InternlmXC-v2 | 8B | @490 | 62.72 | 81.28 | 72.00 | 66.72 | 84.08 | 75.40 | 70.80 | 86.56 | 78.68 |
| Specialist Chart Models | | | | | | | | | | | |
| Pix2Struct | 282M | Ada. | 30.08 | 76.88 | 53.48 | 31.68 | 78.40 | 55.04 | 37.28 | 81.12 | 59.20 |
| Matcha | 282M | Ada. | 37.12 | 86.64 | 61.88 | 39.84 | 87.36 | 63.60 | 43.52 | 88.56 | 66.04 |
| UniChart | 201M | @960 | 34.64 | 83.28 | 58.96 | 36.48 | 84.24 | 60.36 | 38.88 | 85.28 | 62.08 |
| Deplot + LLaVA-v1.6 | 282M+13B | Ada. | 53.44 | 87.68 | 70.56 | 56.80 | 88.48 | 72.64 | 60.64 | 90.08 | 75.36 |
| Chart MLLMs | | | | | | | | | | | |
| ChartVLM | 13B | Ada. | 42.08 | 82.48 | 62.28 | 43.84 | 82.88 | 63.36 | 46.00 | 83.28 | 64.64 |
| OneChart | 125M+13B | @1024 | 54.48 | 87.12 | 70.80 | 57.60 | 87.84 | 72.72 | 62.00 | 88.64 | 75.32 |
| ChartLlama | 13B | @336 | 58.40 | 93.12 | 75.76 | 61.20 | 93.60 | 77.40 | 63.52 | 94.00 | 78.76 |
| ChartGemma+PoT | 3B | @448 | 67.84 | 85.28 | 76.56 | 68.64 | 85.84 | 77.24 | 69.84 | 86.32 | 78.08 |
| TinyChart | 3B | @768† | 58.72 | 94.88 | 76.80 | 62.56 | 95.28 | 78.92 | 67.04 | 96.16 | 81.60 |
| ChartAst | 13B | @448 | 64.88 | 93.12 | 79.00 | 66.24 | 93.84 | 80.04 | 67.44 | 94.32 | 80.88 |
| TinyChart+PoT | 3B | @768† | 70.24 | 90.72 | 80.48 | 71.20 | 91.44 | 81.32 | 72.40 | 92.56 | 82.48 |
| ChartMoE (Ours) | 8B | @490 | 71.36 | 91.04 | 81.20 | 75.12 | 92.48 | 83.80 | 78.16 | 93.68 | 85.92 |
| ChartMoE+PoT (Ours) | 8B | @490 | 78.32 | 90.96 | 84.64 | 80.16 | 92.32 | 86.24 | 82.08 | 93.60 | 87.84 |

all open-sourced models. With better numerical extraction abilities, PoT can significantly enhance our ChartMoE's question-answering performance.

# 4 EXPERIMENT

## 4.1 IMPLEMENTATION DETAILS

During the alignment stage, we train the connector parameters and keep the visual encoder and LLM parameters frozen. In the supervised fine-tuning stage, we continue training the MoE connector while employing LoRA to update the LLM parameters. All training processes are conducted on 4 × A100-40G GPUs. You can refer to Appendix A.2 for more details.

## 4.2 EVALUATION METRICS

**ChartQA** (Masry et al., 2022) test split consists of 1,250 questions in both human and augmented parts. The charts are three common chart types and are sourced from the real world. It features a variety of human-crafted questions and answers to evaluate models' understanding, reasoning, and data extraction skills. ChartQA adopts relaxed accuracy, which is highlighted shortcomings by recent studies (Chen et al., 2024; Xu et al., 2023), such as simplistic string matching and direct float conversion. Therefore, we improve it by 1) using regular expression matching to extract number values, 2) optimizing string matching for short answers, and 3) demonstrating model performance under various relaxed margins. We adopt it for all experiment results.

**ChartBench** (Xu et al., 2023) focuses on charts without data point annotations. It includes a broader range of chart types, with 9 main categories and 42 subcategories, each containing 50 charts. Chart-Bench focuses on extracting numerical values, posing a greater challenge as models cannot depend on OCR for precise answers. It adopts *Acc+* for judgments and relaxed accuracy for NQA tasks. The benchmark proposes to extract number values by LLMs first, which is omitted for the stratifying instruction-following ability of ChartMoE.

**ChartFC** (Akhtar et al., 2023a) & **ChartCheck** (Akhtar et al., 2023b) adopt accuracy to verify whether the claim aligns with the input chart, marking a significant advancement in chart recognition and reasoning abilities. This identifies the potential hallucinations in chart-related contexts. The ChartFC test set has 1,591 questions, and the ChartCheck test set has two splits, containing 937 questions and 981 questions.

## 4.3 COMPARATIVE MODELS

**General MLLMs.** We compare PaliGemma (Beyer et al., 2024), LLaVA-v1.5 (Liu et al., 2023d) with an MLP connector, Qwen-VL (Bai et al., 2023b) with a Qformer (Li et al., 2023) connector,

Table 3: The zero-shot performance on **ChartBench**. No methods are fine-tuned on the trainset for fairness. We exclude PoT because ChartBench mainly assesses numerical extraction accuracy without math calculation.

| Models | Regular Type | | | | Extra Type | | | | | | | ALL |
| --- | --- | --- | --- | --- | --- | --- | --- | --- | --- | --- | --- | --- |
| | Line | Bar | Pie | Avg. | Area | Box | Radar | Scatter | Node | Comb. | Avg. | |
| General MLLMs | | | | | | | | | | | | |
| LLaVA-v1.5 | 29.12 | 21.26 | 17.28 | 22.10 | 21.73 | 20.94 | 27.50 | 23.47 | 36.80 | 24.30 | 24.96 | 23.38 |
| Qwen-VL | 38.00 | 20.71 | 38.24 | 29.46 | 28.83 | 24.17 | 35.00 | 19.50 | 18.50 | 25.50 | 26.56 | 28.18 |
| DocOwl-v1.5 | 49.60 | 31.69 | 31.54 | 35.68 | 12.27 | 23.33 | 22.50 | 36.13 | 29.60 | 38.80 | 27.38 | 32.05 |
| Mini-Gemini | 34.88 | 36.12 | 40.40 | 36.77 | 31.20 | 23.33 | 30.60 | 35.20 | 43.60 | 27.90 | 30.61 | 34.37 |
| InternlmXC-v2 | 68.16 | 48.74 | 56.60 | 54.50 | 27.47 | 25.33 | 40.10 | 52.93 | 50.40 | 46.20 | 39.72 | 48.41 |
| Specialist Chart Models | | | | | | | | | | | | |
| Pix2Struct | 2.56 | 2.37 | 1.90 | 2.33 | 0.13 | 0.13 | 4.60 | 0.67 | 0.40 | 3.20 | 2.93 | 2.16 |
| Matcha | 6.80 | 5.05 | 3.60 | 5.18 | 0.27 | 1.60 | 6.20 | 3.46 | 5.40 | 4.80 | 5.81 | 4.84 |
| UniChart | 7.04 | 5.35 | 4.30 | 5.55 | 3.86 | 4.80 | 11.60 | 5.06 | 15.80 | 9.60 | 8.30 | 6.78 |
| Deplot+LLaVA-v1.6 | 31.20 | 26.46 | 24.00 | 27.09 | 21.34 | 13.34 | 24.00 | 41.34 | 42.00 | 31.00 | 31.57 | 27.62 |
| Chart MLLMs | | | | | | | | | | | | |
| ChartVLM | 21.92 | 14.16 | 10.50 | 15.16 | 7.47 | 7.87 | 8.00 | 7.87 | 5.40 | 10.50 | 8.38 | 11.96 |
| ChartLlama | 26.80 | 18.83 | 20.80 | 20.99 | 14.27 | 12.00 | 24.30 | 27.73 | 26.20 | 25.80 | 21.71 | 21.31 |
| TinyChart | 32.40 | 25.81 | 22.50 | 26.71 | 10.13 | 14.80 | 13.40 | 28.14 | 10.80 | 21.60 | 22.56 | 22.51 |
| OneChart | 41.28 | 30.28 | 29.60 | 32.65 | 19.07 | 13.20 | 24.60 | 38.53 | 34.80 | 27.90 | 31.91 | 29.93 |
| ChartGemma | 50.48 | 38.21 | 32.10 | 39.89 | 28.27 | 24.13 | 28.10 | 48.00 | 41.80 | 43.40 | 42.47 | 38.46 |
| ChartMoE (Ours) | 71.44 | 51.57 | 52.80 | 56.31 | 38.40 | 24.13 | 40.20 | 62.67 | 58.00 | 49.20 | 55.58 | 51.67 |

DocOwl-v1.5 (Hu et al., 2024) that employs multi-level image resolution and token convolution techniques, and the current open-source SOTA, InternlmXC-v2 (Dong et al., 2024).

**Specialist Chart Models.** Previous works specifically design models and algorithms for chart question answering. We compare Pix2Struct (Lee et al., 2023), Matcha (Liu et al., 2023b), UniChart (Masry et al., 2023), and Deplot (Liu et al., 2023a). Notably, Deplot fails to handle questions in arbitrary formats, so we extract table information with Deplot and use LLaVA-v1.6 to answer the questions.

**Chart MLLMs.** Chart-oriented MLLMs are the promising direction for utilizing prior knowledge of LLMs. ChartLLaMA (Han et al., 2023) proposes to generate high-quality instruction data to improve chart question-answering capabilities. ChartAst (Meng et al., 2024) suggests aligning the connector with chart-table pairs before supervised fine-tuning. ChartVLM (Xia et al., 2024) uses different decoders to handle different questions based on their difficulty. ChartInstruct (Masry et al., 2024a) conducts large-scale chart instruction tuning based on general MLLMs. OneChart (Chen et al., 2024) converts the chart to the table with a dedicated decoder and uses LLMs to answer questions. ChartGemma (Masry et al., 2024b) proposes more instruction data and achieves efficient chart reasoning based on SigLIP (Zhai et al., 2023) and Gemma-2B (Team et al., 2024a). Tiny-Chart (Zhang et al., 2024) adopts token merge to reduce visual tokens and enable high-resolution chart input.

## 4.4 MAIN RESULTS

**ChartQA.** Tab.2 presents detailed comparisons of ChartMoE on ChartQA. ChartMoE significantly improves the baseline (InternlmXC-v2) performance (72.00% vs. 84.64%, +12.64%↑ in Acc.@0.05). Compared to previous SOTA (TinyChart+PoT @768 pixel), ChartMoE consistently surpasses it across all metrics. The PoT effectively enhances the mathematical reasoning capabilities, which is a common shortfall in current MLLMs. ChartMoE integrates better with PoT, indicating that it accurately extracts fundamental elements from charts. ChartMoE shows more significant improvement on *Human* part, especially after incorporating PoT, where the questions are more computationally complex and challenging. Notably, our error analysis in the *Augmented* part reveals that many errors stem from limitations of the evaluation criteria, i.e., string matching. For instance, it is marked incorrect if the prediction is *It is between 2003 and 2005* and the ground truth is *(2003, 2005)*. Forcing performance improvement may lead to model overfitting.

**ChartBench.** Tab. 3 presents detailed comparisons of ChartMoE on ChartBench. None of the models, including our ChartMoE, undergo supervised fine-tuning on the ChartBench trainset to ensure fair experimental comparison. Chart-specific models typically underperform due to limited generalization, which fails to manage the annotated charts effectively ($< 10\%$). Deplot shows a distinct advantage over these types of models (27.62%) with the assistance of LLaVA-v1.6. The

baseline (InternlmXC-v2) demonstrates strong generalization on ChartBench (48.41%), which may benefit from pre-training instructions designed for unannotated charts. Without additional design, ChartMoE improves the baseline performance comprehensively (48.41% vs. 51.67%), especially on extra chart types (39.72% vs. 55.58%, +15.86%↑).

**ChartFC & ChartCheck.** Tab. 4 compares ChartMoE on the synthetic ChartFC and real-world ChartCheck. ChartMoE consistently outperforms SOTA (e.g., ChartGemma +4.4%↑ on ChartFC) and significantly improves the performance compared to InternlmXC-v2 (+6.83%↑ and +8.76%↑ on ChartCheck T1 and T2, respectively). Note that this is implemented without using training data for supervised fine-tuning, demonstrating ChartMoE's strong generalization capabilities.

| Models | ChartFC | ChartCheck | |
|---|---|---|---|
| | | T1 | T2 |
| PaliGemma | 58.26 | 67.34 | 68.50 |
| LLaVA-v1.5† | 61.28 | 70.22 | 70.03 |
| InternlmXC-v2 | 65.93 | 72.04 | 70.44 |
| ChartInstruct-LLama2 | 69.57 | 70.11 | 68.80 |
| ChartInstruct-FlanT5XL | 70.27 | 72.03 | 73.80 |
| ChartGemma | 70.33 | 71.50 | 74.31 |
| ChartMoE (Ours) | **74.73** | **78.87** | **79.20** |

Table 4: The accuracy performance on *ChartFC* and *ChartCheck*. †: tuning with ChartGemma instructions.

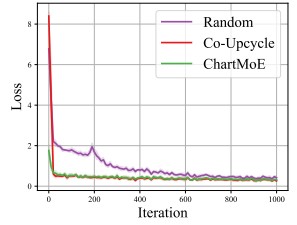

Figure 4: Training loss of different initialization.

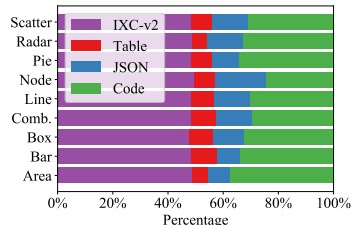

Figure 5: Top-2 selected expert distribution on ChartBench.

# 5 FURTHER STUDY

## 5.1 MODEL ARCHITECTURE ABLATION

We investigate the impact of three factors on our MoE connector: the number of experts, the number of activated experts, and the expert initialization manner. All the experiments are conducted with ChartQA training data and evaluated on ChartQA test split with relax accuracy metric.

**Effect of the Expert Initialization Manner.** The initialization strategy plays a crucial role in determining the performance of the MoE connector. Effective initialization is essential to ensure that each expert performs its designated function optimally. As illustrated in Tab. 5 row 1-3, we explore the impact of 3 initialization strategies for the MoE connector. Random initialization serves as a baseline but struggles with convergence (refer to Fig. 4), resulting in a suboptimal accuracy of 73.20% at Acc.@0.05. Following CuMo (Li et al., 2024), we employ the Co-Upcycle strategy by replicating the *table-JSON-code* aligned connector for all experts. Given the same starting point, this approach lacks expert diversity, which limits its effectiveness, resulting in an accuracy of 77.48% at Acc.@0.05. In contrast, our initialization assigns distinct parameters to each expert. This tailored approach enables each expert to capitalize on its specific strengths, resulting in the highest performance, achieving 78.76% in Acc.@0.05.

**Effect of Number of Experts and Activated Experts.** As shown in rows 3-4 of Tab. 5, we compare ChartMoE configurations with 4 and 8 experts, keeping 2 experts activated. The 8 experts are initialized in pairs using the 4 methods illustrated in Fig. 2c. ChartMoE achieves 78.76% in Acc.@0.05 with 4 experts, which is slightly higher than the 78.60% achieved with 8 experts, showing a marginal increase of +0.16%. In rows 4-5, we compare the performance of configurations with 2 and 4 activated experts, finding similar results: 78.60% vs. 78.64% in Acc.@0.05. This analysis suggests that merely increasing the number of experts or the activation of experts does not guarantee improved performance. The configuration with 4 experts and 2 activated experts effectively balances complexity and performance, making it a suitable choice for ChartMoE.

## 5.2 TRAINING STRATEGY ABLATION

We analyze the impact of the training strategy across alignment and supervised fine-tuning stages. We use InternlmXC-v2 with ChartQA fine-tuning as our baseline, maintaining the same hyperparameters as the chart-specific annealing tuning stage.

**Effect of Alignment Strategy.** As shown in rows 1-3 of Tab. 6, translating the chart image into structural text formats such as table, JSON, and code during the alignment stage significantly enhances performance in downstream chart understanding tasks. After applying *table-JSON-*

Table 5: Ablation study on ChartMoE architecture w.r.t. the total / activated / initialization of connector experts. All experiments are conducted on ChartQA.

| Total Experts | Activated Experts | Experts Initialization | Relax Acc @0.05 | | |
|---|---|---|---|---|---|
| | | | Human | Aug. | Avg. |
| 4 | 2 | Random Init. | 59.68 | 86.72 | 73.20 |
| 4 | 2 | Random Align | 62.32 | 88.88 | 75.60 |
| 4 | 2 | Co-Upcycle Init. | 64.96 | 90.00 | 77.48 |
| 4 | 2 | Diversely Align | **67.92** | 89.60 | **78.76** |
| 8 | 2 | Diversely Align | 67.20 | **90.00** | 78.60 |
| 8 | 4 | Diversely Align | 67.68 | 89.60 | 78.64 |

Table 6: Ablation study on the proposed training strategy and connector architecture on the alignment, high-quality knowledge learning, and chart-specific anneal tuning stages.

| ChartMoE Recipe | Relax Acc @0.05 | | |
|---|---|---|---|
| | Human | Aug. | Avg. |
| Baseline: InternlmXC-v2 + ChartQA | 63.68 | 87.68 | 75.68 |
| + *table-JSON-code* Aligned Connector | 64.24 | 90.16 | 77.20 |
| + *Top2-in-4* ChartMoE Connector | 67.92 | 89.60 | 78.76 |
| + *MMC* High-Quality Knowledge Learning | 67.84 | 90.24 | 79.04 |
| + *ChartGemma* Instructions | **71.36** | **91.04** | **81.20** |

Table 7: Ablation study on the expert of MoE connector. We ignore the gating network and adopt specific expert output.

| Connector | Relax Acc @0.05 | | |
|---|---|---|---|
| | Human | Aug. | Avg. |
| Expert 0 (Vanilla) | **69.76** | **89.84** | **79.80** |
| Expert 1 (Table) | 63.60 | 89.12 | 76.36 |
| Expert 2 (JSON) | 60.64 | 82.48 | 71.56 |
| Expert 3 (Code) | 66.88 | 89.36 | 78.12 |

Table 8: Ablation study on alignment pre-training tasks. We adopt different alignment tasks for baseline (linear connector) and further conduct supervised fine-tuning on the ChartQA train set.

| Alignment | *w/o* ChartQA SFT | | | *w/i* ChartQA SFT | | |
|---|---|---|---|---|---|---|
| | Human | Aug. | Avg. | Human | Aug. | Avg. |
| Vanilla | **62.72** | **81.28** | **72.00** | 63.68 | 87.68 | 75.68 |
| Table | 51.28 | 71.76 | 61.52 | 63.92 | 89.28 | 76.60 |
| JSON | 44.40 | 65.12 | 54.76 | **64.88** | 89.84 | **77.36** |
| Code | 50.16 | 71.20 | 60.68 | 64.24 | **90.16** | 77.20 |

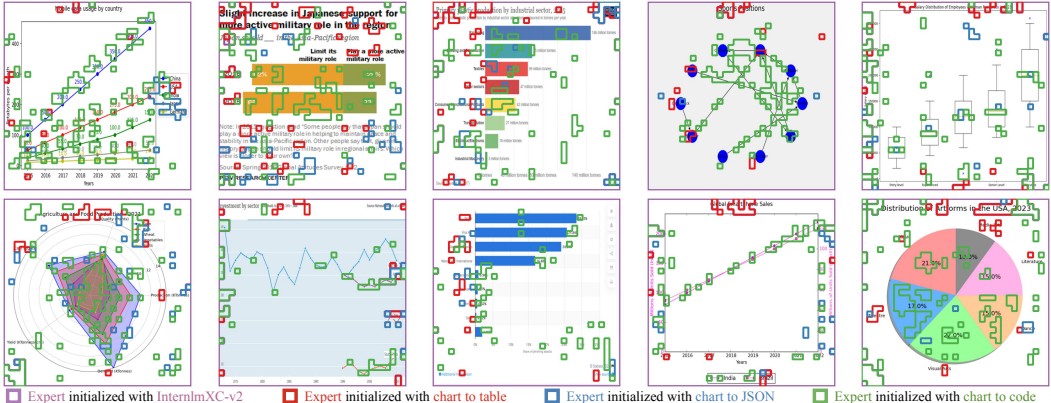

Figure 6: Visualizations of top-1 expert selection. Only the boundaries of the merged tokens are plotted.

*code* alignment, the model achieves 77.20% in Acc.@0.05, representing a notable improvement (+1.52%↑). When combined with our proposed MoE connector, performance further increases to 78.76%, a total gain of +3.08%↑ in Acc.@0.05.

**Effect of Supervise Fine-Tuning Strategy.** As shown in rows 4-5 of Tab. 6, we divide the supervised fine-tuning stage into two phases. By incorporating high-quality knowledge learning using the MMC dataset, ChartMoE achieves 79.04% Acc.@0.05, reflecting a 3.36% improvement. In the chart-specific annealing tuning phase, we introduce ChartGemma data to enhance the model's reasoning and PoT capabilities, leading the model to peak performance (81.20%, +5.52%↑).

## 5.3 IN-DEPTH ANALYSIS

**Effect of the Each Expert.** To explore the role of each expert in ChartMoE, we bypass the gating network and manually select the output of specific experts. As shown in Tab. 7, E0 performs the best (79.80%), which is consistent with the distribution in Fig. 5. However, this doesn't mean other experts lack relevance, which may offer crucial insights at key moments (Fig. 6).

**Effect of Alignment Task.** As shown in Tab. 8, we explore various alignment tasks based on the linear connector. After alignment, the performance on ChartQA declines compared to the baseline. However, the aligned model exhibits a substantial improvement after supervised fine-tuning on the ChartQA train split, which is consistent with previous observations (Meng et al., 2024; Yan et al., 2024). Specifically, the JSON and code tasks exhibit remarkable improvement over the table.

**Expert Distribution Visualization.** As shown in Fig. 5& 6, we visualize the expert distribution in the MoE connector on the ChartBench test set. We designate the vanilla connector as E0, while E1-3 corresponds to connectors aligned with tables, JSON, and code. As depicted in Fig. 5, the

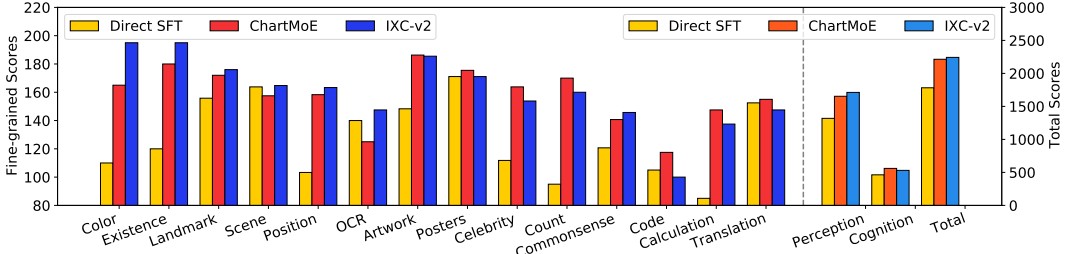

Figure 7: The performance on the general VQA tasks (MME Fu et al. (2023)). With supervised fine-tuning on extensive chart-structured data, the directly tuned IXC-v2 shows a significant performance drop, while ChartMoE maintains a satisfying performance by keeping the vanilla connector as the expert in MoE.

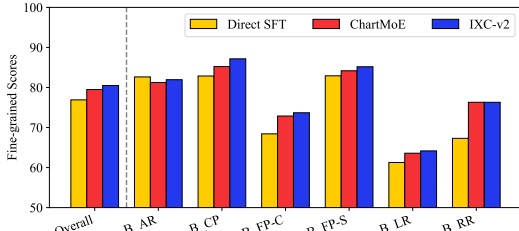

Figure 8: The performance on the general VQA tasks (MMBench Liu et al. (2023e)). Please refer to its paper for each task's details. The observations and conclusions are consistent with the MME benchmark.

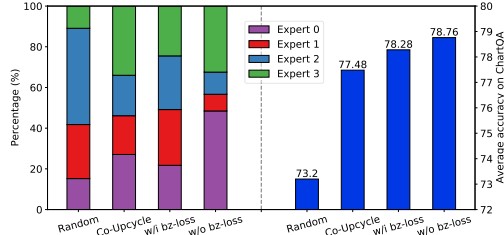

Figure 9: The performance with/without bz-loss on ChartQA. Left: The bz-loss leads to more even expert selections. Right: A more balanced distribution does not yield better performance.

trend is consistent across different chart types, with E0 and E3 being the most frequently selected connectors. The expert selection shows no extreme bias, as even the least chosen, E1, accounts for over 10%. We further visualize the expert selection for each image token, revealing the preferences of each expert. As shown in Fig. 6, E0 is the primary choice for background tokens, explaining its dominance in Fig. 5. E1 and E2 are more frequently chosen by tokens from titles, axis labels, or legends, as these elements are commonly found in tables and JSON files. ChartMoE tends to use E3 to focus on the data points and visual elements within the chart, e.g., data points on the line, digital text, and edges in a node chart. These components are essential for accurately re-drawing the charts.

**Performance on General Tasks.** While ChartMoE is designed to enhance chart understanding, it does not compromise other capabilities, e.g., instruction following and object recognition. In Fig. 7 & 8, we show the comparisons of directly fine-tuned InternlmXC-v2 (short for Direct SFT) with data from Tab. 1 and the baseline (short for IXC-v2) on general benchmarks (Fu et al., 2023; Liu et al., 2023e). The direct SFT model shows diminished general capabilities. In contrast, ChartMoE preserves it nearly intact by retaining the original connector as one of its experts.

**Effect of Balance Loss in MoE.** The standard MoE (Zoph et al., 2022) employs balanced loss and router z-loss (short for bz-loss) to prevent certain experts from dominating the model training. In Fig. 9, we compare the effects of with and without bz-loss. While bz-loss promotes a more equitable selection of experts, it fails to enhance ChartMoE's performance further. As shown in Fig. 6, the expert initialization in ChartMoE results in each expert having its own preference for visual token selection (refer to Appendix B for detail). Consequently, the bz-loss might hinder the model's convergence to the optimal point because the distribution of visual tokens is inherently imbalanced.

# 6 CONCLUSION

We introduce ChartMoE, a multi-task aligned and instruction-tuned MLLM designed for complex chart understanding and reasoning. We replace the linear connector with the MoE architecture and initialize each expert with parameters derived from different alignment tasks. We further present the ChartMoE-Align dataset, a synthetic collection of nearly 1 million table-json-code-chart quadruples, to facilitate alignment training across different experts. This approach preserves the strengths of each alignment task, ensuring efficient training and superior model performance. ChartMoE outperforms the previous state-of-the-art on several benchmarks by a large margin and excels in real-world applications such as chart question answering, translation, and editing. Please refer to Appendix A.3 for the reproducibility statement.

# 7 ACKNOWLEDGMENTS

This work was supported by the National Key R&D Program of China (2022YFB4701400/470140 2), SSTIC Grant(KJZD20230923115106012, KJZD20230923114916032, GJHZ202402181136040 08), and Beijing Key Lab of Networked Multimedia.

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

# A ADDITIONAL EXPERIMENTAL SETTINGS AND RESULTS

## A.1 TOP-2 EXPERTS DISTRIBUTION

Our ChartMoE employs MoE connector expert parameters initialized with various alignment tasks. To investigate the impact of these initialization methods on model performance, we present the comparisons in Tab. 6& 7& 8 and Fig. 4&5. For a deeper analysis, we explore how different initialization methods affect expert selection. As shown in Fig. 10, both random initialization and co-upcycle result in a more uniform distribution of experts. However, this uniformity does not inherently lead to improved performance or interpretability, possibly due to insufficient differentiation among the experts. In contrast, our ChartMoE clearly prefers specialized roles, as illustrated in Fig. 6& 11& 12.

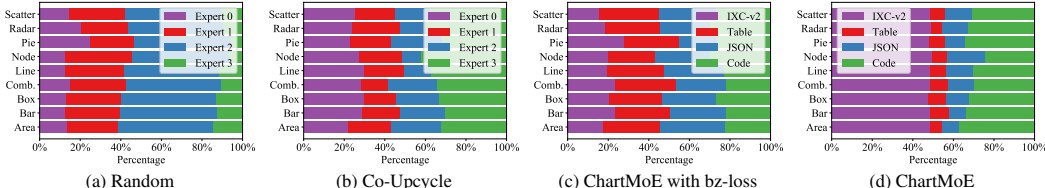

|  (a) Random | (b) Co-Upcycle | (c) ChartMoE with bz-loss | (d) ChartMoE |

Figure 10: The distribution of Top-2 experts after supervised fine-tuning with three expert initialization methods. We calculate the proportion of the top 2 experts selected by the router on the ChartBench.

## A.2 SUMMARY OF HYPERPARAMETER SETTINGS

The training process of our ChartMoE is structured into three distinct phases: Alignment Pre-training, High-Quality Knowledge Learning, and Chart-Specific Annealing Tuning. Table 9 provides a comprehensive overview of the hyperparameter configurations employed during each training stage. It should be noted that using flash-attention (Dao, 2024) can effectively reduce GPU hours.

Table 9: Training hyperparameters of ChartMoE for all stages.

| Configuration | Alignment Pre-training | High-Quality Knowledge Learning | Chart Specific Annealing Tuning |
|---|---|---|---|
| Connector Initialization | InternlmXC-v2 | Table&JSON&Code Experts + InternlmXC-v2 | ChartMoE 2nd-stage |
| LLM Training | Freeze | LoRA | LoRA |
| Image Resolution | 490 | 490 | 490 |
| ViT Sequence Length | 1225 | 1225 | 1225 |
| Optimizer | AdamW | AdamW | AdamW |
| Optimizer Hyperparameter | $\beta_1 = 0.9, \beta_2 = 0.95, \epsilon = 1e^{-8}$ | $\beta_1 = 0.9, \beta_2 = 0.95, \epsilon = 1e^{-8}$ | $\beta_1 = 0.9, \beta_2 = 0.95, \epsilon = 1e^{-8}$ |
| Peak Learning Rate | $5e^{-5}$ | $5e^{-5}$ | $1e^{-5}$ |
| Learning Rate Schedule | cosine decay | cosine decay | cosine decay |
| Weight Decay | 0.1 | 0.1 | 0.1 |
| Gradient Clip | 1.0 | 1.0 | 1.0 |
| Warm-up Ratio | 0.01 | 0.01 | 0.01 |
| Global Batch Size | 256 | 64 | 64 |
| Gradient Acc. | 16 | 8 | 8 |
| Numerical Precision | bfloat16 | bfloat16 | bfloat16 |
| Optimizer Sharding | ✓ | ✓ | ✓ |
| Gradient Sharding | ✓ | ✓ | ✓ |
| Parameter Sharding | ✗ | ✗ | ✗ |
| Activation Checkpointing | ✓ | ✓ | ✓ |
| GPU Hours (A100-40G) | 210 | 100 | 56 |

## A.3 REPRODUCIBILITY STATEMENT

We have included the architecture of ChartMoE in Section 3.1 and the complete training procedure in Section 3.2 and Section 3.3. The training data recipe is listed in Tab. 1 in detail. Hyper-parameter settings are shown in Appendix A.2. We also introduce the generation pipeline for ChartMoE-Align in Section 3.2, and some detailed examples in Appendix C. Furthermore, our ChartMoE-Align dataset and checkpoints of ChartMoE will be released soon on GitHub and Huggingface.

# B ADDITIONAL VSUALIZATIONS OF TOP-1 EXPERT SELECTION

In this section, we randomly sampled images from natural image datasets (LLaVA-CC3M (Liu et al., 2023d)) and chart datasets (ChartQA (Masry et al., 2022), ChartBench (Xu et al., 2023)) to illustrate ChartMoE's token selection preferences. As shown in Fig. 11, the vanilla expert focuses more on the background, the table expert concentrates on details such as the boundary between the background and the subject, the JSON expert focuses on textures (e.g., maps and objects), and the code expert specializes in curves and trends (e.g., logos and text). Fig. 12 further demonstrates that while the vanilla expert continues to attend to background tokens, critical visual elements are handled by the aligned experts, with the code expert being notably more prominent.

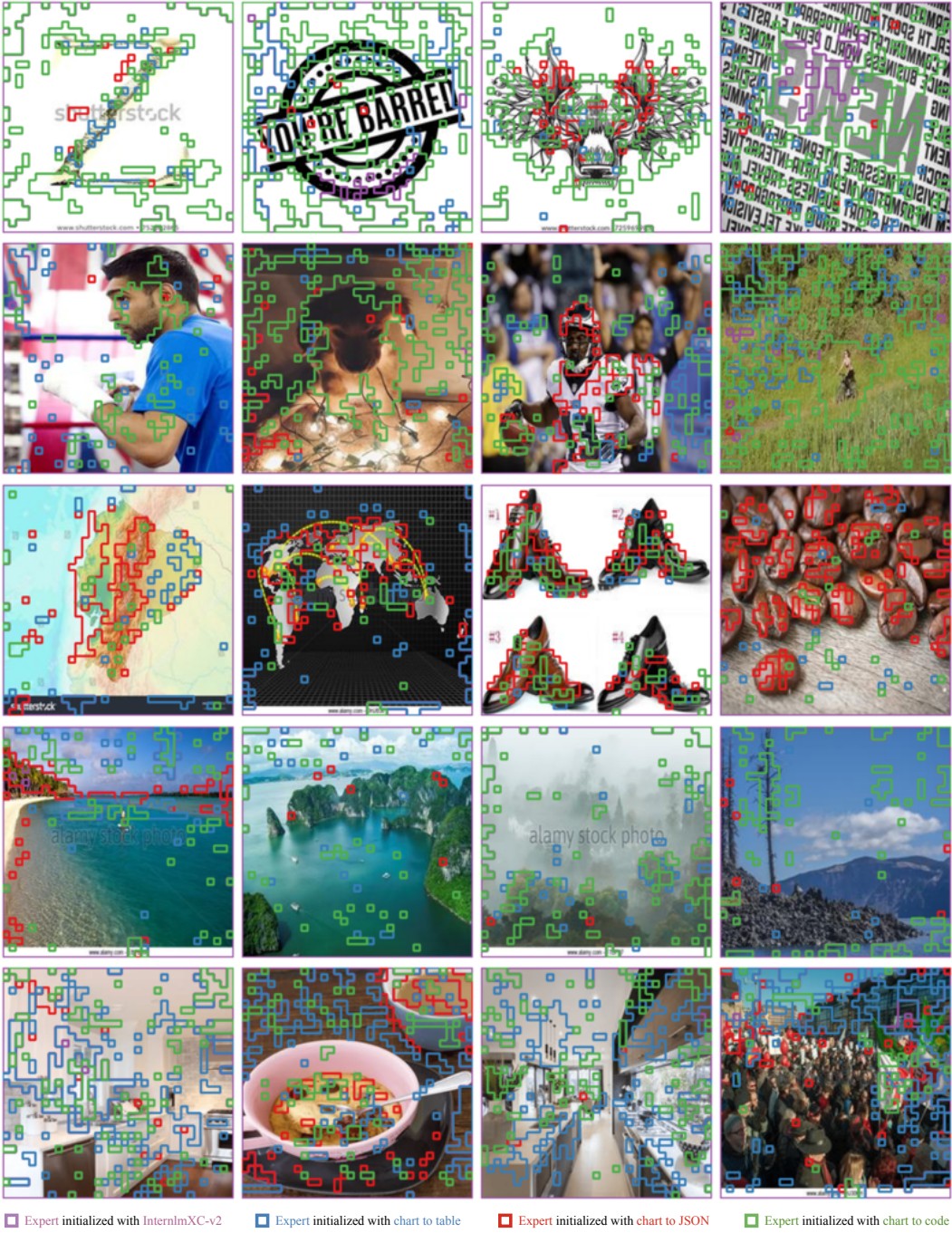

□ Expert initialized with InternlmXC-v2    □ Expert initialized with chart to table    □ Expert initialized with chart to JSON    □ Expert initialized with chart to code

Figure 11: More visualizations of top-1 expert selection on *general images* randomly sampled from LLaVA-CC3M. These examples show the selection preferences of different experts in ChartMoE.

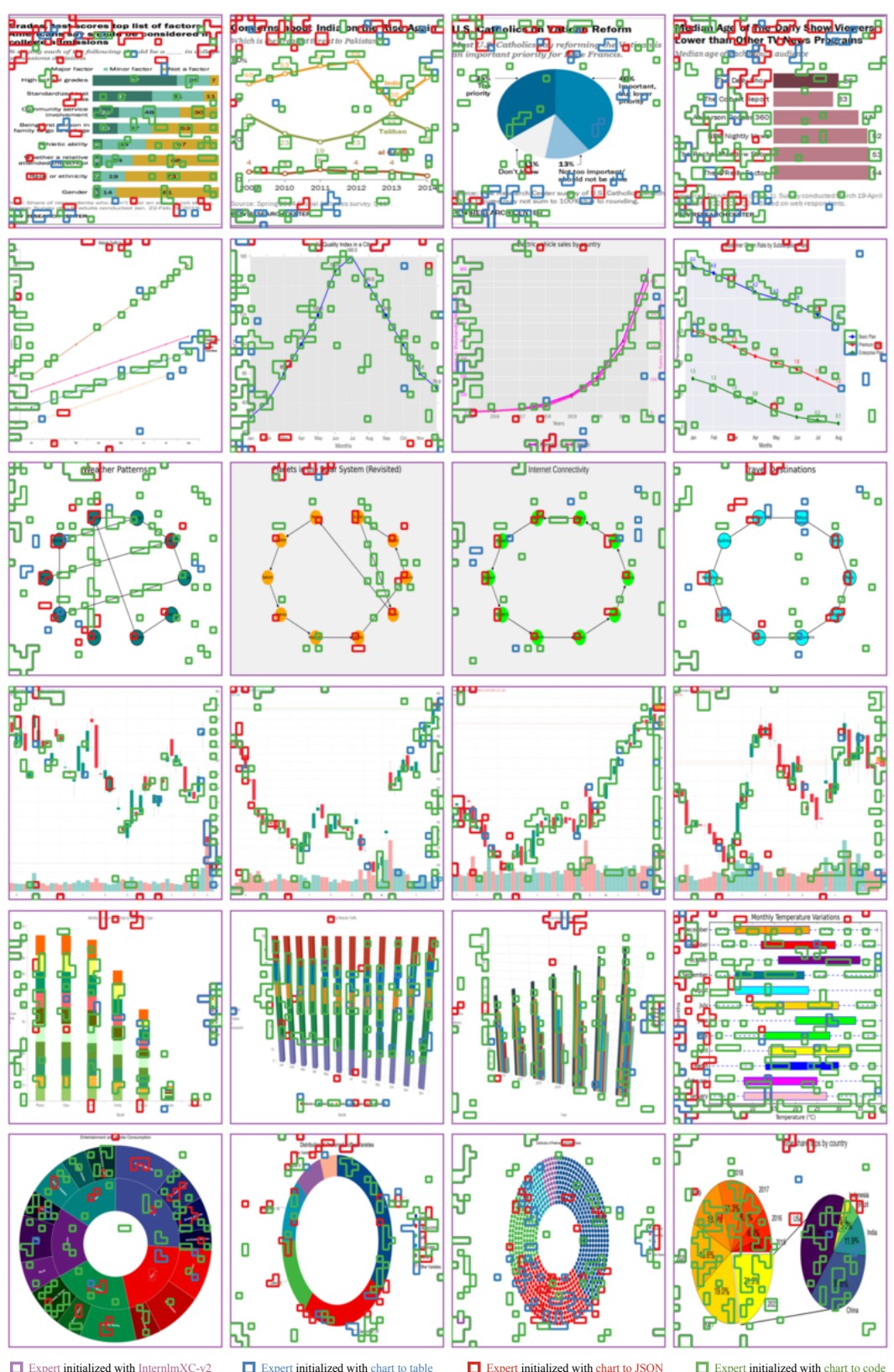

Figure 12: More visualizations of top-1 expert selection on ***chart images***. The vanilla expert primarily handles background tokens, and the chart visual markers are handled by other experts.

## C   DETAILS OF CHARTMOE-ALIGN

### C.1   OVERVIEW

ChartMoE-Align is a dataset we introduced for different experts aligning pretraining. It consists of nearly 1 million Chart Table JSON Code quadruples and supports three alignment tasks: Chart to Table, Chart to JSON, and Chart to Code. Unlike other chart datasets, ChartMoE-Align focuses solely on these three fundamental alignment tasks without considering the diversity of instruction tasks.

### C.2   TABLE DATA COLLECTION

We primarily collect table data from three sources: the ChartQA training set (Masry et al., 2022), the PlotQA training set (Methani et al., 2020), and ChartY provided by OneChart (Chen et al., 2024).

**ChartQA** includes 18.3K real-world charts and provides accompanying meta tables. While the charts are of high quality and manually curated, they lack fine-grained attribute annotations and executable plotting code. As a result, we only retained the tables from ChartQA in CSV format.

**PlotQA** comprises 157K charts, primarily focusing on three common types: line, bar, and pie charts. These charts are generated using Python code with limited formatting and style diversity. Consequently, we did not utilize the charts from PlotQA but retained its 157K tables. These tables originate from sources like World Bank Open Data, Open Government Data, and the Global Terrorism Database, covering statistics on various indicators such as fertility rates, rainfall, coal production, and more across years, countries, and districts.

**ChartY** is a chart dataset containing 2.7M charts in both Chinese and English proposed by OneChart. Notably, ChartY also includes charts from ChartQA and PlotQA, which we filtered out in ChartMoE-Align. Additionally, ChartY primarily consists of common chart types such as line, bar, and pie charts (or their combinations) and suffers from significant data imbalance. To address this, we sampled a subset to ensure a roughly equal number of charts for each type. As the tables in ChartY are mainly generated by GPT-3.5 based on templates, we ultimately retained 763K samples from this source.

### C.3   PAIR DATA CONSTRUCTION

**JSON** provides a structured format distinct from CSV, designed to retain chart attributes beyond numerical data, such as title position, font size, element colors, legend styles, and more. We adopt the template provided by ChartReformer (Yan et al., 2024) and further enhance it. We add chart type-agnostic attributes like title position and gridlines. For chart type-specific attributes, we aim to remain consistent with ChartReformer's definitions while accommodating all chart types present in ChartMoE-Align. With this framework, we generate corresponding JSON files for all tables. To extract chart type-specific attributes, we fine-tune a Deplot (Liu et al., 2023a) model, leveraging the original chart to extract their properties. Missing attributes are filled in using random sampling to ensure completeness.

**Code** refers to Python scripts based on *matplotlib* for rendering the charts. Leveraging the rich attributes defined in the JSON, the code is designed to faithfully represent every attribute to ensure diversity in the resulting charts. During generation, we explicitly specify all default parameters, such as the hexadecimal color codes for each line/bar, default font sizes, text positions, etc. We provide basic code templates for type-agnostic attributes. For type-specific attributes, rules are used to automatically generate the corresponding code.

**Chart** is produced by executing the generated code. Given the number of table, JSON, and code pairs, we filter out any quadruples with execution errors or warnings during the chart generation process, retaining only valid and error-free samples.

**Instruction**. Considering the alignment task, we directly employ several templated questions to define the Chart-to-X tasks (X is the ground truth). Ultimately, each quadruple corresponds to three QA pairs. Note that ChartMoE-Align only serves for alignment training to initialize different expert projectors, thus emphasizing the diversity of charts and aligned modalities. To improve model performance and instruction-following, we still require more diverse instructions for supervised fine-tuning to update the MoE connector and LLM.

### C.4   QUALITY CONTROL

We first remove all duplicate entries from the meta table and then eliminate quadruples that cause errors or warnings during rendering. To further assess the quality of ChartMoE-Align, we randomly sample 200 quadruples and ask GPT-4o and annotation experts (with at least three experts reviewing each quadruple) to evaluate the clarity and readability of the charts, as well as the alignment between the charts and table/JSON/code, scoring them as 0 or 1. The results show that nearly all charts are clear, unambiguous, and free from obstructions (GPT-4o: 96.5%, Experts: 99%). Over 90% of the pairs are matching and suitable for instruction tuning (GPT-4o: 91%, Experts: 94.5%).

## C.5 EXAMPLE VISUALIZATION

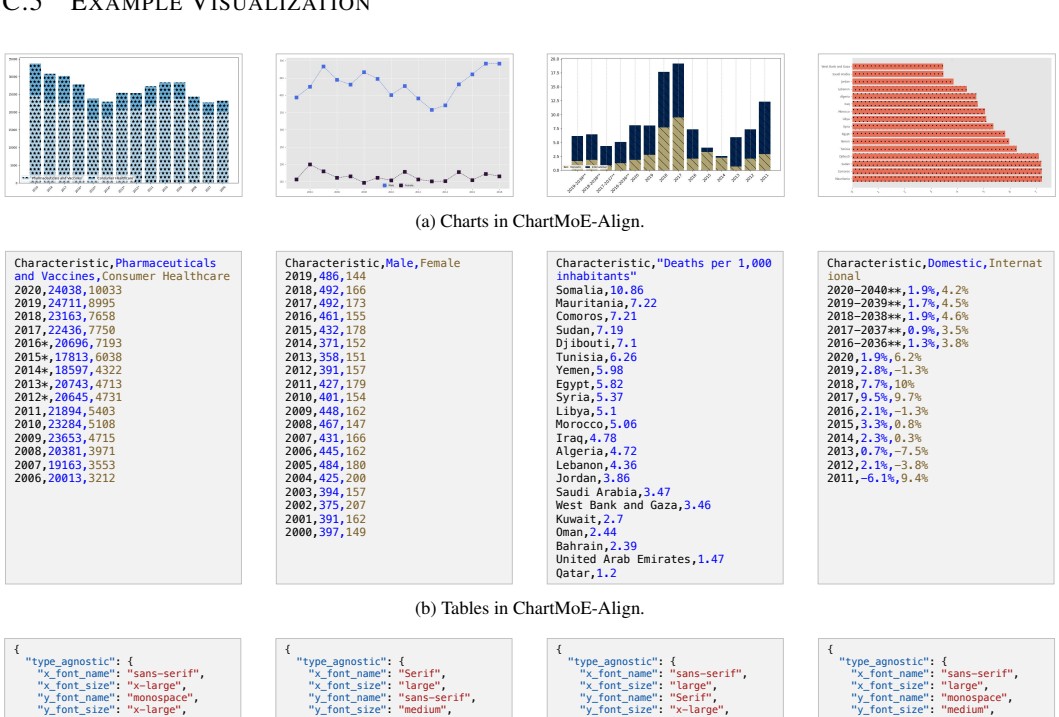

Figure 13: Detailed Examples in ChartMoE-Align. Each quadruple contains the chart, table, JSON and code.

# D  FURTHER DISCUSSION

## D.1  CONTRIBUTION OF CHARTMOE

Some prior work, such as MoE-LLaVA (Lin et al., 2024), DeepSeek-VL (Lu et al., 2024), and CuMo (Li et al., 2024), has employed MoE architectures in MLLMs. However, these approaches all apply MoE to LLMs or ViTs to increase model capacity, introducing a large number of learnable parameters to boost performance. In contrast, our ChartMoE introduces several distinctive innovations:

**1) Motivation**: Our goal is not to expand model capacity but to enhance the model's chart comprehension through alignment tasks while preserving performance on other general tasks. Hence, we retain the original connector parameters as one expert initialization manner.

**2) Initialization**: Unlike previous methods that rely on random or co-upcycle initialization, we leverage multiple alignment tasks for expert (connector) initialization. This approach enables ChartMoE to exhibit remarkable interpretability (Fig. 6 & 11 & 12).

**3) Complexity**: We are the first to apply MoE exclusively to the MLP connector (projector) in LLaVA-like MLLMs. In ChartMoE (based on InternlmXC-v2), the MoE architecture introduces minimal additional parameters (model size 8.364B → 8.427B, + 63M↑ only) and training complexity (Fig. 4). It also shows negligible impact on inference speed (0.945 → 0.952 seconds per QA on ChartQA test set) and peak memory usage (23.72 GB → 23.86 GB, *fp16* on A100-40G GPU).

## D.2  CHARTMOE BASED ON OTHER MLLMs

Our ChartMoE is based on InterlmXC-v2, but our proposals (MoE connector, diverse alignment, etc.) are general approaches. Therefore, we use 10% of the alignment data (Tab. 1) and the ChartQA training data to train our proposals based on LLaVA-v1.5-7B to further demonstrate their effectiveness. As shown in Tab. 10, our proposals significantly improve the base model. This is partly because LLaVA is trained with fewer chart data, leading to a lower baseline, and also indicates that the additional alignment data greatly enhances chart understanding.

Table 10: Performance comparison on ChartQA with LLaVA-v1.5-7B as base MLLM.

| Models | Relax Acc @0.05 | | | Relax Acc @0.10 | | | Relax Acc @0.20 | | |
|---|---|---|---|---|---|---|---|---|---|
| | Human | Aug | Avg | Human | Aug | Avg | Human | Aug | Avg |
| LLaVA-v1.5-7B | 7.60 | 7.36 | 7.48 | 7.92 | 8.08 | 8.00 | 9.04 | 9.52 | 9.28 |
| LLaVA-v1.5-7B + ChartQA | 6.08 | 23.04 | 14.56 | 8.24 | 32.96 | 20.60 | 10.32 | 42.16 | 26.24 |
| LLaVA-v1.5-7B + ChartMoE | 18.13 | 32.11 | 25.12 | 20.20 | 42.32 | 31.36 | 24.24 | 52.12 | 38.18 |

## D.3  PERFORMANCE ON CHARTQA

In Tab. 2, our ChartMoE significantly outperforms SOTA. However, some models perform better than ours on the *Augment* part of the ChartQA test set. Given that the *Augment* part of ChartQA is considerably easier than the *Human* part, we conduct a more detailed analysis. We analyze the performance of various models on numeric (*Human*: 43%, *Augment*: 39%) and non-numeric (*Human*: 57%, *Augment*: 61%) questions. As shown in Tab. 11, ChartMoE excels in all subcategories except for non-numeric questions in the *Augment* part. We find that ChartMoE's errors primarily occur in string-matching tasks. For instance, a prediction of *It is between 2003 and 2005* is marked incorrect if the ground truth is *(2003, 2005)*. High accuracy in this category may indicate overfitting instead.

Table 11: Fine-grained performance comparison on ChartQA with error margin 5%.

| Method | Human | | | Augment | | | Acc |
|---|---|---|---|---|---|---|---|
| | Numeric | Non-Numeric | Avg | Numeric | Non-Numeric | Avg | |
| TinyChart | 58.52% | 58.03% | 58.24% | 92.43% | **96.25**% | 94.32% | 76.28% |
| ChartAst | 67.04% | 65.35% | 66.08% | 93.20% | **93.07**% | 93.12% | 79.00% |
| ChartMoE (Ours) | 73.89% | 75.49% | 74.80% | 93.20% | **90.98**% | 91.84% | 84.64% |

## D.4  LIMITATIONS

ChartMoE has two limitations: 1) Dependency on alignment tasks. ChartMoE requires chart-Table/JSON/Code alignment tasks for initialization. Non-chart multimodal tasks need new alignment designs to initialize MoE experts. 2) Limited flexibility. Modifying the projector into a multi-expert architecture makes ChartMoE non-plug-and-play like LoRA. We are required to retrain the router network when new experts are coming.

# E    CAPABILITY COMPARISONS ON CHART EDITING AND HIGHLIGHTING

ChartMoE, leveraging carefully designed alignment training, excels in accurately extracting chart values and attributes, facilitating precise image reconstruction, chart element editing, and highlighting. In this section, we provide a visual comparison with the SOTA MLLMs, Qwen-2.5 (Alibaba, 2024), and GPT-4O (OpenAI, 2024). While all three models perform the required instructions effectively, ChartMoE demonstrates superior accuracy in value and attribute extraction.

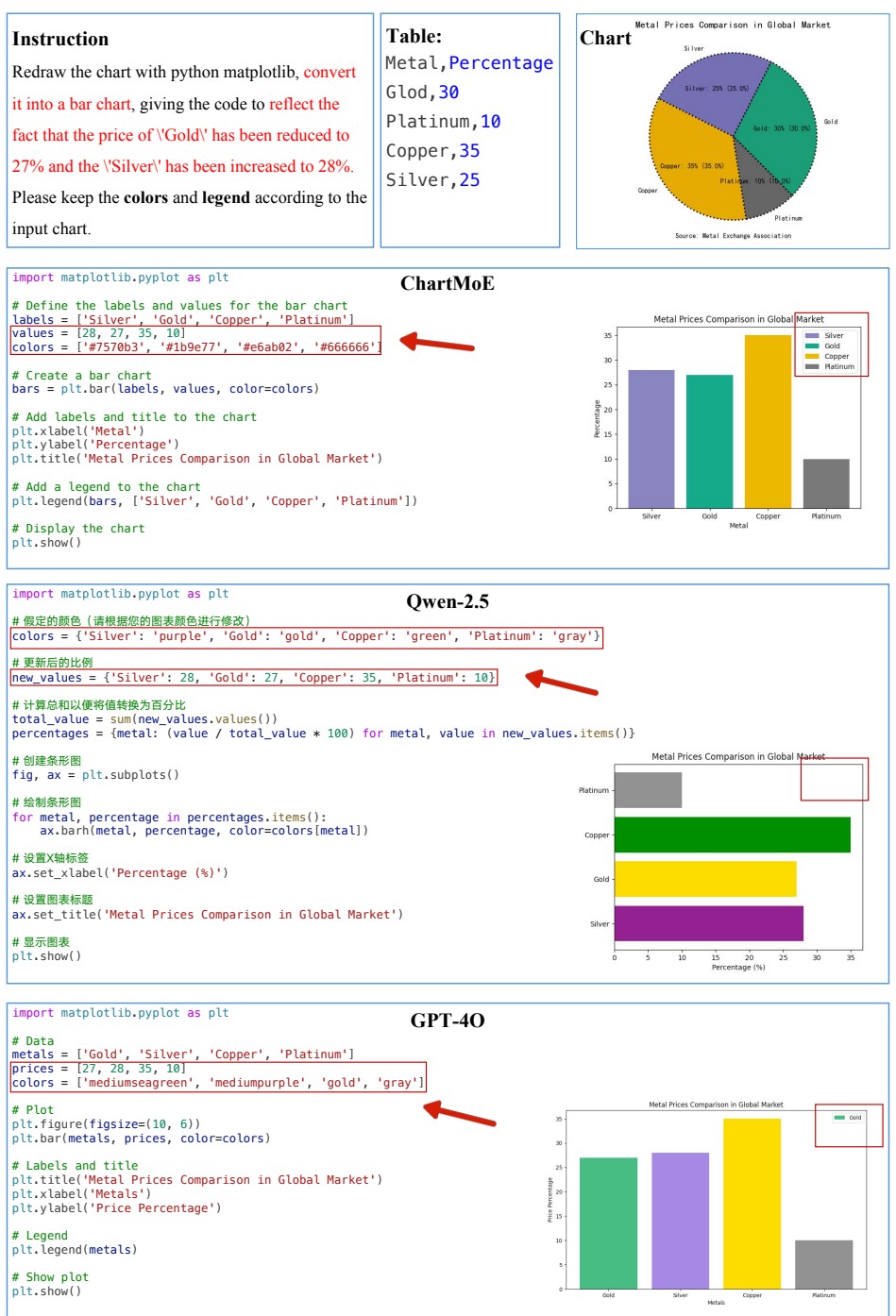

Figure 14: Chat demo involves *modifying chart types and values*. All models successfully convert the chart type, but only ChartMoE handles the legend correctly. No model makes errors in this task due to the simplicity of the values and the presence of data point labels.

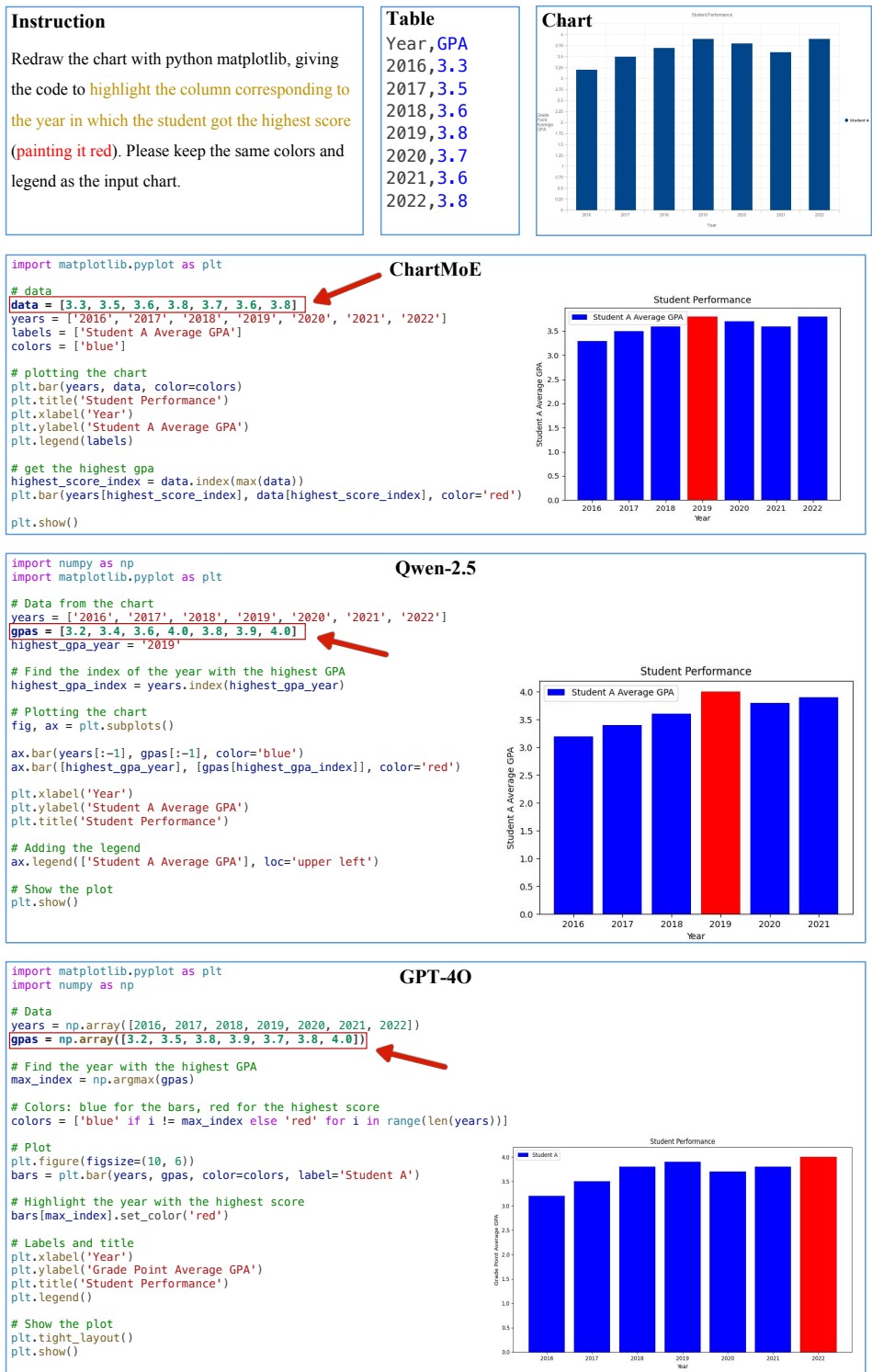

Figure 15: Chat demo involves modifying *chart editing*. The bar chart is without labeled data points, and all methods provide reasonable and executable drawing code. Qwen-2.5 directly identifies the highest element, while the other two methods make it by code. Note that ChartMoE delivers the most accurate values, thanks to extensive alignment training and proposed MoE architecture.

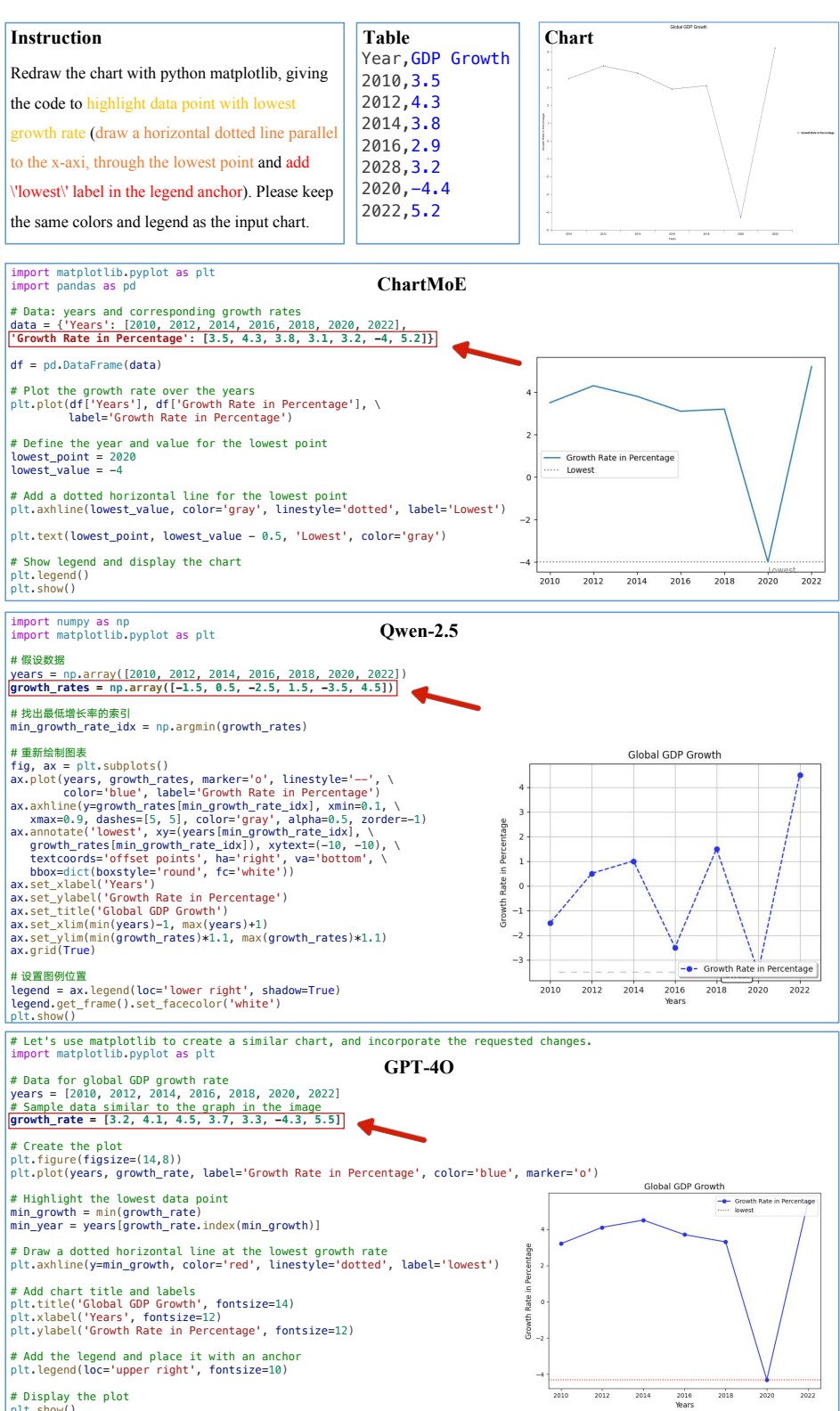

Figure 16: Chat demo involves modifying ***chart editing***. The line chart is without labeled data points, and all methods provide reasonable and executable drawing code. The values extracted by all models differ from the ground truth, but both ChartMoE and GPT-4O captured the correct data trends. Additionally, ChartMoE successfully completed all the editing tasks specified in the instructions.