# OpenReview forum: "ChartMoE: Mixture of Diversely Aligned Expert Connector for Chart Understanding"
_ICLR.cc/2025/Conference — ICLR 2025 Oral_

### Official Review · Reviewer_Ka6r · 2024-11-03

**Soundness:** 3
**Presentation:** 3
**Contribution:** 3
**Rating:** 8
**Confidence:** 3

**Summary:**

This paper proposed ChartMoE, which employs the Mixture of Expert (MoE) architecture to replace the traditional linear projector to bridge the modality gap.

**Strengths:**

1. First introduce MoE MLLM for chart task, the model architechture design which using MoE in connector is novel.
2. Detailed experiments and analysis, extensivequantitativeandqualitativestudies.
3. Introduced a large dataset for chart pre-train data.

**Weaknesses:**

1. limmited innovation: MoE on multimodal language models has been explored in other domains. There's a trade-off between the performance gain and the increase of model parameters and inference time.
2. The contributions of MoE module is unclear, more theoretical analysis is needed such as the knowledge in the routing to differnernt experts.

**Questions:**

Have you ever tried training on your alignment data with random initialization of expert parameters and balanced loss from scratch? It will better prove the significance of this work.

---

> ### Author Response · Authors · 2024-11-20
>
> Thanks for your detail review!
>
> ---
>
> ### **Q1: About Novelty and Tradd-off of MoE**
>
> > MoE on multimodal language models has been explored in other domains. There's a trade-off between the performance gain and the increase of model parameters and inference time.
>
> We would sufficiently highlight the distinctions between ChartMoE and other MoE models in MLLMs. Some prior work, such as MoE-LLaVA[1], DeepSeek-VL[2], and CuMo[3], has employed MoE architectures in MLLMs. However, these approaches all apply MoE to **LLMs** or **ViTs** to increase model capacity, introducing a large number of learnable parameters to boost performance. In contrast, our ChartMoE introduces several distinctive innovations:
>
> 1. **Motivation**: Our goal is not to expand model capacity but to enhance the model's chart comprehension through alignment tasks while preserving performance on other general tasks. Hence, we retain the original connector parameters as one expert initialization manner.
>
> 2. **Initialization**: Unlike previous methods that rely on random or co-upcycle initialization, we leverage diverse alignment tasks for expert (connector) initialization. This approach enables ChartMoE to exhibit remarkable interpretability (Fig.6,11,12). From the perspective of visual tokens, experts aligned with Table and JSON focus more on regions with text, such as titles and legends, excelling in tasks similar to OCR. In contrast, the Code expert focuses more on data points and trend directions, excelling in visual logical reasoning and overall analysis.
>
> 3. **Complexity**: We are the **first** to apply MoE exclusively to **the MLP connector (projector)** in LLaVA-like MLLMs. In ChartMoE (based on InternlmXC-v2), the MoE architecture introduces minimal additional parameters (model size **8.364B $\rightarrow$ 8.427B, + 63M$\uparrow$** only) and training complexity (Fig.4). It also shows negligible impact on inference speed (**0.945 $\rightarrow$ 0.952 seconds per QA** on ChartQA test set) and peak memory usage (**23.72 GB $\rightarrow$ 23.86 GB**, *fp16* on A100-40G GPU).
>
> We have added this discussion to Appendix D.1 to make our paper more comprehensive. We hope our explanation helps you reassess the novelty of ChartMoE!
>
> ---
>
> ### **Q2: More Theoretical Analysis of MoE**
> Our motivation for using the MoE connector is primarily based on the following two points:
>
> 1. Different structured texts contain varying core information and information volumes, leading the aligned experts to focus on different regions of the charts.
>
> 2. We aim to enhance chart understanding capabilities while preserving the original model's performance on general tasks.
>
> The visualization results (Figures 6, 11, 12) show that the vanilla expert tends to handle background tokens, the table&Json experts focus more on text and textures, and the code expert pays more attention to data trends. This insight led us to remove the bz-loss from the standard MoE architecture, as the visual tokens are not evenly distributed, and forcing an equal workload for each expert is not optimal. We will explore more in-depth theoretical analysis in our future work.
>
> ---
>
> ### **Q3: More Ablation Study**
>
> > Have you ever tried training on your alignment data with random initialization of expert parameters and balanced loss from scratch? It will better prove the significance of this work.
>
> Thank you for providing an important baseline setting in the ablation study! We have added the results for this setting in Table 5. *Align* refers to using ChartAlign for alignment, while *Init.* refers to random initialization without alignment. The experimental results demonstrate the significance of ChartMoE.
>
> ---
>
> We are glad to see your recognition of our contribution. We would be happy to answer any remaining questions or concerns!
>
> **Reference:**
>
> [1] Moe-llava: Mixture of Experts for Large Vision-Language Models, Arxiv 2024.
>
> [2] Deepseek-vl: Towards Real-world Vision-Language Understanding, Arxiv 2024.
>
> [3] CuMo: Scaling Multimodal LLM with Co-Upcycled Mixture-of-Experts, Arxiv 2024.

---

> ### Author Response · Authors · 2024-11-24
> **Awaiting Your Confirmation**
>
> Dear Reviewer Ka6r,
>
> With the discussion phase ending in **3 days**, we would greatly appreciate any further guidance or confirmation from your side. We wanted to follow up on our previous response to your valuable suggestions. Your insights are very important to us, and we are eager to ensure that we have addressed all concerns appropriately.
>
> Warm Regards, :-)

---

> > ### Comment · Reviewer_Ka6r · 2024-11-27
> > **Thanks for the response**
> >
> > I have checked the response, and I realized that the authors have addressed my most concerns. Therefore, I slightly raise my rating.
> >
> > Thanks.

---

### Official Review · Reviewer_WXLS · 2024-11-03

**Soundness:** 4
**Presentation:** 3
**Contribution:** 3
**Rating:** 8
**Confidence:** 4

**Summary:**

The paper introduces ChartMoE, a multi-task aligned and instruction-tuned MLLM designed for complex chart understanding and reasoning. The key contribution is the replacement of the traditional linear connector with the Mixture of Expert (MoE) architecture, which improves chart understanding by bridging the modality gap. Additionally, a new dataset called ChartMoE-Align is introduced, containing nearly 1 million chart-table-JSON-code quadruples for alignment training. The proposed three-stage training paradigm and high-quality knowledge learning approach result in significantly improved performance compared to the previous state-of-the-art on various benchmarks.

**Strengths:**

This paper incorporates a Mixture of Expert architecture to bridge the gap between charts and language models and offers a valuable insight for the expert initialization manner. The creation of the ChartMoE-Align dataset with nearly 1 million chart-table-JSON-code quadruplets is a significant contribution to the field, allowing for detailed and meticulous chart alignment pre-training.

The paper is clear and well written in general, is well motivated, and comes with an extensive and comprehensive ablation study.

**Weaknesses:**

The paper focuses on conducting experiments on a single Vision Encoder and Large Language Model, which limits the generalizability of the proposed method. It would be beneficial to test the effectiveness of ChartMoE on a diverse set of MLLMs to ensure its applicability across different models and scenarios.

The paper does not thoroughly discuss potential limitations or challenges that may arise when implementing ChartMoE in practical applications.

**Questions:**

1. In line 369, the error analysis suggests that many errors stem from the limitations of the evaluation metric, namely string matching. Are other comparison models also limited by this?
2. Can you provide more detailed explanations of the data construction process, such as how code templates for different types of charts were obtained? Were they manually constructed?

---

> ### Author Response · Authors · 2024-11-20
>
> Thank you for your review and detailed comments! Your suggestion is valuable for us to make this paper comprehensive.
>
> ---
>
> ### **Q1: ChartMoE Base on Other MLLMs**
>
> Thank you for your very reasonable suggestions. ChartMoE can be applied to all LLaVA-like MLLMs. Therefore, we conducted experiments based on LLaVA-v1.5-7B, using 10% alignment data (Tab. 1) and ChartQA training data, and the results are shown in the table below.
>
> |        Models       | Human |  Aug  | Acc @0.05 | Human |  Aug  | Acc @0.10 | Human |  Aug  | Acc @0.20 |
> |:-------------------:|:-----:|:-----:|:---------:|:-----:|:-----:|:---------:|:-----:|:-----:|:---------:|
> |   LLaVA-v1.5-7B   |  7.60 |  7.36 |    7.48   |  7.92 |  8.08 |    8.00   |  9.04 |  9.52 |    9.28   |
> |    LLaVA-v1.5-7B + ChartQA |  6.08 | 23.04 |   14.56   |  8.24 | 32.96 |   20.60   | 10.32 | 42.16 |   26.24   |
> | LLaVA-v1.5-7B + ChartMoE | 18.13 | 32.11 |   25.12   | 20.20 | 42.32 |   31.36   | 24.24 | 52.12 |   38.18   |
>
> As shown in the table, our proposals significantly improve the base model. This is partly because LLaVA is trained with fewer chart data, leading to a lower baseline performance, and also indicates that the additional alignment data and MoE connector greatly enhance chart understanding. We have added this discussion to Appendix D.2 to make our paper more comprehensive.
>
> ---
>
> ### **Q2: About Limitations of ChartMoE**
>
> ChartMoE has two limitations:
>
> 1. Dependency on alignment tasks. ChartMoE requires chart-Table/JSON/Code alignment tasks for initialization. Non-chart multimodal tasks need new alignment designs to initialize MoE experts.
>
> 2. Limited flexibility. Modifying the projector into a multi-expert architecture makes ChartMoE non-plug-and-play like LoRA. We are required to retrain the router network when new experts are coming.
>
> ---
>
> ### **Q3: About Metric on ChartQA**
>
> > Are other comparison models also limited by ChartQA evaluation metric？
>
> Yes, other methods are also affected by the evaluation criteria. We report results that we have reproduced ourselves or re-evaluated using new criteria based on the inference result files provided by the original authors.
>
> ---
>
> ### **Q4: About ChartMoE-Align Details**
>
> We have updated Appendix C, where we detail the construction process of ChartMoE-Align. We provide code templates for chart type-agnostic attrbutions and adopt ruled-based methods to fit different chart types. Please refer to Appendix C for details. We will release all sources of ChartMoE-Align and ChartMoE for research purposes.
>
> ---
>
> We hope to have addressed your concerns and are thankful for your recognition of our contributions!

---

> ### Author Response · Authors · 2024-11-24
> **Awaiting Your Confirmation**
>
> Dear Reviewer WXLS,
>
> With the discussion phase ending in **3 days**, we would greatly appreciate any further guidance or confirmation from your side. We wanted to follow up on our previous response to your valuable suggestions. Your insights are very important to us, and we are eager to ensure that we have addressed all concerns appropriately.
>
> Warm Regards, :-)

---

> > ### Comment · Reviewer_WXLS · 2024-11-26
> > **Official Comment by Reviewer WXLS**
> >
> > Thank you for your thoughtful response. I appreciate your efforts, and my score remains equally positive. Best regards.

---

### Official Review · Reviewer_6BC6 · 2024-11-04

**Soundness:** 2
**Presentation:** 3
**Contribution:** 3
**Rating:** 5
**Confidence:** 4

**Summary:**

This work introduces ChartMoE, a multimodal large language model  that enhances automatic chart understanding through a MoE architecture. Unlike traditional linear connectors, ChartMoE uses diverse expert connectors aligned with specific tasks (chart-table, chart-JSON, and chart-code) to bridge the gap between visual encoders and large language models. The paper also presents ChartMoE-Align, a dataset with nearly 1 million chart-table-JSON-code quadruples for training.

**Strengths:**

1. ChartMoE’s use of task-specific expert connectors in a Mixture of Experts (MoE) framework provides a  solution to multimodal chart understanding.

2. ChartMoE-Align, a large-scale dataset with varied chart alignments (table, JSON, code).

3. The three-stage training paradigm increass its accuracy in extracting and interpreting numerical data.

**Weaknesses:**

1. The use of MOE in MLLMs is not particularly novel, as several prior works have already explored MoE structures to enhance model performance. From an innovation standpoint, this reliance on MoE does not introduce a distinctly new approach and could be considered a weakness in terms of contribution.

2. The multi-expert structure, along with the diverse alignment tasks, adds significant complexity to ChartMoE’s architecture. Also, the training data, being mostly synthetic, might limit the model’s ability.

**Questions:**

see above

---

> ### Author Response · Authors · 2024-11-20
>
> Thanks for your review and detailed comments! We hope the following discussion can address your concerns!
>
> ---
>
> ### **Q1: About the Novelty of MoE in MLLMs**
>
> > The use of MoE in MLLMs is not particularly novel, as several prior works have already explored MoE structures to enhance model performance. From an innovation standpoint, this reliance on MoE does not introduce a distinctly new approach and could be considered a weakness in terms of contribution.
>
> We would sufficiently highlight the distinctions between ChartMoE and other MoE MLLMs. Some prior work, such as MoE-LLaVA[1], DeepSeek-VL[2], and CuMo[3], has employed MoE architectures in MLLMs. However, these approaches all apply MoE to **LLMs** or **ViTs** to increase model capacity, introducing a large number of learnable parameters to boost performance. In contrast, our ChartMoE introduces several distinctive innovations:
>
> 1. **Motivation**: Our goal is not to expand model capacity but to enhance the model's chart comprehension through alignment tasks while preserving performance on other general tasks. Hence, we retain the original connector parameters as one expert initialization manner.
>
> 2. **Initialization**: Unlike previous methods that rely on random or co-upcycle initialization, we leverage diverse alignment tasks for expert (connector) initialization. This approach enables ChartMoE to exhibit remarkable interpretability (Fig.6,11,12). From a visual token perspective, experts aligned with Table and JSON prioritize text regions like titles and legends, excelling in OCR-like tasks. In contrast, the Code expert emphasizes data points and trend directions, demonstrating strengths in visual logical reasoning and overall analysis.
>
> 3. **Complexity**: We are the **first** to apply MoE exclusively to **the MLP connector (projector)** in LLaVA-like MLLMs. In ChartMoE (based on InternlmXC-v2), the MoE architecture introduces minimal additional parameters (model size **8.364B $\rightarrow$ 8.427B, + 63M$\uparrow$** only) and training complexity (Fig.4). It also shows negligible impact on inference speed (**0.945 $\rightarrow$ 0.952 seconds per QA** on ChartQA test set) and peak memory usage (**23.72 GB $\rightarrow$ 23.86 GB**, *fp16* on A100-40G GPU).
>
> We have added this discussion to Appendix D.1 to make our paper more comprehensive. We hope our explanation helps you re-assess the novelty of ChartMoE. Thank you!
>
> ---
>
> ### **Q2: MoE Architecture and Differentiated Alignment Tasks Add Complexity**
>
> > The multi-expert structure, along with the diverse alignment tasks, adds significant complexity to ChartMoE’s architecture.
>
> 1. **Model Complexity**. In **Q1**, we introduced the minimal increase in model parameters, training difficulty, inference speed, and inference memory requirements introduced by the MoE connector (projector). This is because we only replace the linear projector with an MoE at the connector position, making the overall changes minimal.
>
> 2. **Training Complexity**. ChartMoE does not significantly increase computation costs. The expert alignment process in ChartMoE uses 800k training data (500k tables for expert 1 / 200k JSONs for expert 2 / 100k codes for expert 3), and the training parameters include only a single MLP expert. The computation cost of the alignment process is equivalent to training an MLP Connector with part of the ChartMoE-Align data (800k), where the difference is the training data is split by task type for 3 stages.
>
> Therefore, ChartMoE introduces almost no additional model complexity or training complexity compared to other MLLMs with LLaVA structures. We have added this discussion to Appendix D.1 to make our paper more comprehensive. We hope our explanation is helpful for you.

---

> ### Author Response · Authors · 2024-11-20
>
> ### **Q3: Synthetic Data Might Limit the Model’s Ability**
>
> > Also, the training data, being mostly synthetic, might limit the model’s ability.
>
> Yes, real data is typically more effective than synthetic data. However, collecting a sufficient number of real-world charts with meta tables is challenging, and obtaining charts with meta JSON and code is even more difficult. Recent studies [4-5] have shown that synthetic data can significantly aid model training. The synthetic data in ChartMoE-Align has also been carefully diversified and quality-controlled.
>
> Additionally, we only use synthetic data for the alignment stages. During the SFT stage for training the MoE router and LLM, we still use real data from MMC, ChartQA, and ChartGemma, which helps ChartMoE achieve SOTA performance on multiple benchmarks. The ablation experiments in Tables 5-8 also demonstrate the effectiveness of synthetic data in ChartMoE training.
>
> ---
>
> We hope to have addressed your concerns.
>
> **Reference:**
>
> [1] Moe-llava: Mixture of Experts for Large Vision-Language Models, Arxiv 2024.
>
> [2] Deepseek-vl: Towards Real-world Vision-Language Understanding, Arxiv 2024.
>
> [3] CuMo: Scaling Multimodal LLM with Co-Upcycled Mixture-of-Experts, Arxiv 2024.
>
> [4] Self-Rewarding Language Models, Arxiv 2024.
>
> [5] Reinforced Self-Training for Language Modeling, Arxiv 2024.

---

> ### Author Response · Authors · 2024-11-24
> **Awaiting Your Confirmation**
>
> Dear Reviewer 6BC6,
>
> With the discussion phase ending in **3 days**, we would greatly appreciate any further guidance or confirmation from your side. We wanted to follow up on our previous response to your valuable suggestions. Your insights are very important to us, and we are eager to ensure that we have addressed all concerns appropriately.
>
> Warm Regards, :-)

---

> > ### Comment · Reviewer_6BC6 · 2024-11-25
> >
> > Dear Author,
> >
> > I have reviewed your rebuttal, and while it has addressed some of my questions, I still have some doubts about the novelty of the work. As a result, I will not be changing my score (I considered increasing it by 0.5, but ICLR does not allow for such increments). If AC decided to accept then I am fine with that too.
> >
> > Thank you for clarifying my questions.

---

> > > ### Author Response · Authors · 2024-11-25
> > >
> > > Dear Reviewer 6BC6,
> > >
> > > Thank you very much for your timely response. We are glad that our reply has addressed some of your concerns!
> > >
> > > If you have a moment, could you share more detailed thoughts on the novelty aspect? We’re looking forward to delving deeper into the application of MoE in Chart MLLM and believe our discussion could spark new ideas for our future work.
> > >
> > > Your support is incredibly valuable to us!
> > >
> > > Warm Regards, :-)

---

### Official Review · Reviewer_jNMd · 2024-11-05

**Soundness:** 4
**Presentation:** 4
**Contribution:** 3
**Rating:** 6
**Confidence:** 2

**Summary:**

The paper introduces ChartMoE, a novel approach that leverages a Mixture of Expert (MoE) architecture to improve automatic chart understanding with Multimodal Large Language Models (MLLMs). ChartMoE addresses the limitations of existing MLLMs by using specialized linear connectors for diverse expert initialization, coupled with a unique dataset, ChartMoE-Align, containing nearly one million chart-related quadruples. This setup significantly enhances data interpretation from charts, pushing the accuracy on the ChartQA benchmark from 80.48% to 84.64%, showcasing a substantial improvement over previous methods.

**Strengths:**

S1. Innovative Methodology: The introduction of the ChartMoE method, which utilizes a Mixture of Experts (MoE) architecture, represents a significant innovation in the field of automatic chart understanding. This approach addresses the modality gap effectively and could potentially set a new direction for future research in multimodal learning systems.

S2. Comprehensive Dataset: The creation of the ChartMoE-Align dataset is commendable. Its large scale and diversity are well-suited for robust pre-training in chart alignment tasks. This dataset not only serves the immediate needs of the study but also provides a valuable resource for the broader research community to explore complex multimodal tasks involving charts and text.

S3. Extensive Experimental Validation: The paper presents extensive experiments that demonstrate the effectiveness of the ChartMoE approach. The thoroughness of these experiments, which include a variety of scenarios and detailed performance metrics, establishes a strong benchmark for future comparative studies.

S4. Clear Writing: The manuscript is exceptionally well-written, providing clear explanations and methodical presentation of the concepts and methodologies involved. This clarity enhances the reader's understanding and appreciation of the work's contributions to the field.

**Weaknesses:**

W1: Details on Dataset Construction

The paper lacks critical details on the dataset construction process. Clarifications are needed regarding the criteria used to select and filter charts for inclusion in the dataset. Specifically, the process for generating meta CSV data via Large Language Models (LLM) requires more transparency.
More details on which LLMs were used and the code templates for different types of charts are missing. Such information is crucial for reproducibility and for understanding the dataset's applicability to other multimodal tasks.
The manuscript should discuss the steps taken to ensure the quality of the data, including any validation mechanisms or controls used during dataset assembly.

W2: Clarification of Experiment Results

The paper briefly mentions that the proposed method shows weaknesses in some settings compared to baselines, as detailed in Tables 2 and 3. However, these points are not adequately addressed or explained. A more thorough analysis of why ChartMoE underperforms in these instances would be valuable for readers and for future improvements to the method.

**Questions:**

Q1: Modality-Specific Contributions

It would be beneficial for the paper to elaborate on the unique contributions of different representations (JSON, code, and chart) in the context of chart-related tasks. Understanding how each representation impacts the model's learning and performance could provide insights into optimizing future models for similar tasks.


Q2: Necessity and Efficiency of Large-Scale Alignment Dataset

The heuristic approach to generating a large-scale dataset raises questions about the efficiency and necessity of such a volume of data. Is there potential to achieve similar performance with a smaller, possibly more curated dataset? This exploration could lead to more resource-efficient training processes and better generalization in practical applications.

---

> ### Author Response · Authors · 2024-11-20
>
> Thank you for your review and detailed comments! We are glad to see your recognition of our contribution. You can find below our detailed response to your questions and concerns. Please let us know if you have any further concerns or suggestions.
>
> ---
>
> ### **Q1: Details on ChartMoE-Align Construction**
>
> Thanks for your suggestions! We have updated Appendix C, where we detail the construction process of ChartMoE-Align. Specifically, ChartMoE-Align uses real-world tables (from ChartQA and PlotQA) and GPT-generated tables (from ChartY provided by OneChart[1]). We further filter the data in ChartY, removing duplicates and maintaining a balance of chart categories. We adopt the JSON template provided by ChartReformer[2] and further categorize attributes into chart type-specific and type-agnostic. During code generation, we use fixed templates for type-agnostic parts and rule-based methods to ensure faithful representation of each attribute for type-specific parts. Finally, we discard all quadruples that produce rendering errors or warnings. To ensure data quality, we randomly sample 200 quadruples and evaluate them using both human and GPT4 judges to assess chart quality and the match between chart, table, JSON, and code. The evaluation results show that ChartMoE-Align is sufficient for the alignment training task. Please refer to the appendix for a better reading experience. Combined with the description in Section 3.2, we hope this addresses your concerns.
>
> ---
>
> ### **Q2: Clarification of Experiment Results**
>
> > The paper briefly mentions that the proposed method shows weaknesses in some settings compared to baselines, as detailed in Tables 2 and 3. However, these points are not adequately addressed or explained. A more thorough analysis of why ChartMoE underperforms in these instances would be valuable for readers and for future improvements to the method.
>
> Thank you for your suggestions! We add a discussion in Appendix D.3 regarding why ChartMoE performs less well than previous SOTA in certain settings. In Tab.1, our ChartMoE significantly outperforms SOTA. However, some models perform better than ours on the *Augment* part of the ChartQA test set. Given that the *Augment* part of ChartQA is considerably easier than the *Human* part, we conduct a more detailed analysis. We analyze the performance of various models on numeric (*Human*: 43%, *Augment*: 39%) and non-numeric (*Human*: 57%, *Augment*: 61%) questions in ChartQA. As shown in the following table, ChartMoE excels in all subcategories except for non-numeric questions in the *Augment* part. We find that ChartMoE's errors primarily occur in string-matching tasks. For instance, the prediction of *It is between 2003 and 2005* is marked incorrect if the ground truth is *(2003, 2005)*. High accuracy in this category may indicate overfitting instead.
>
> |    Method    | Number | Non-Number |  Human | Number | Non-Number | Augment |   Acc  |
> |:------------:|:------:|:----------:|:------:|:------:|:----------:|:-------:|:------:|
> |   TinyChart  | 58.52% |   58.03%   | 58.24% | 92.43% |   **96.25**%   |  94.32% | 76.28% |
> |   ChartAst   | 67.04% |   65.35%   | 66.08% | 93.20% |   **93.07**%   |  93.12% | 79.00% |
> | ChartMoE-PoT | 73.89% |   75.49%   | 74.80% | 93.20% |   **90.98**%   |  91.84% | 84.64% |
>
> Table 3 shows the results on ChartBench, where ChartMoE does not achieve the best performance in the Pie and Box subcategories. This is because:
>
> 1. The sample sizes for these two subcategories are small, which may introduce random bias.
>
> 2. A more detailed error analysis reveals that most of the errors occur in the *Chart Type* tasks for these subcategories (e.g., <u>Q: This is a combination chart, not a pie chart.</u>). We further instruct ChartMoE to provide reasons for its judgments (<u>A: This graph contains multiple charts, including pie, so it can be considered either a combination chart or a pie chart, depending on what you are asking about.</u>). Therefore, the performance gap is due to the ambiguity in these cases combined with the small sample size of these subcategories, making it non-representative.
>
> ---

---

> ### Author Response · Authors · 2024-11-20
>
> ### **Q3: Modality-Specific Contributions**
>
> > It would be beneficial for the paper to elaborate on the unique contributions of different representations (JSON, code, and chart) in the context of chart-related tasks. Understanding how each representation impacts the model's learning and performance could provide insights into optimizing future models for similar tasks.
>
> Thanks for your suggestions. To demonstrate the individual contributions of each component, we conduct two ablation studies:
>
> 1. Table 7 shows the performance of ChartMoE when experts are manually activated (without considering the router). The vanilla expert demonstrates the best performance, which aligns with its activation frequency and the number of tokens it processes. The code expert also performs well, indicating that this modality brings significant benefits to the system.
>
> 2. Table 8 presents the results when using a single linear projector instead of the MoE architecture, trained with different alignment tasks. Without supervised fine-tuning (SFT) on ChartQA, each alignment task degrades the performance, likely due to the significant differences between alignment tasks and QA tasks. After SFT on ChartQA, all three experts outperform the vanilla expert, demonstrating the effectiveness of the alignment tasks.
>
> We illustrate each expert's preference for selecting visual tokens in Figures 6, 11, and 12. From the perspective of visual tokens, experts aligned with Table and JSON focus more on regions with text, such as titles and legends, excelling in tasks similar to OCR. In contrast, the Code expert focuses more on data points and trend directions, excelling in visual logical reasoning and overall analysis.
>
> We hope these experimental results are helpful to you.
>
> ---
>
> ### **Q4: Necessity and Efficiency of Large-Scale Alignment Dataset**
>
> > Can a smaller but more carefully designed dataset achieve similar alignment effects?
>
> **We believe it can**. However:
>
> 1. Collecting real, finely annotated Table-Chart pairs requires more resources, and collecting Table-JSON-Code-Chart quadruples is even more challenging.
>
> 2. The alignment task is inherently a relatively simple and fixed task. Therefore, we believe that using the cleaned and quality-checked ChartMoE-Align dataset is sufficient. This is reflected in the data ratios shown in Table 1, where the loss for certain modalities saturates after training for a specific number of iterations. As a result, we do not use the full amount of data for each modality.
>
> ---
>
> We hope to have addressed your concerns!
>
> **Reference:**
>
> [1] OneChart: Purify the Chart Structural Extraction via One Auxiliary Token, Arxiv 2024.
>
> [2] ChartReformer: Natural Language-Driven Chart Image Editing, Arxiv 2024.

---

> > ### Comment · Reviewer_jNMd · 2024-12-03
> > **Reply to Authors**
> >
> > Thanks for the detailed response from the authors. I think your reply has addressed most of my concerns. I will maintain my positive score.

---

> ### Author Response · Authors · 2024-11-24
> **Awaiting Your Confirmation**
>
> Dear Reviewer jNMd,
>
> With the discussion phase ending in **3 days**, we would greatly appreciate any further guidance or confirmation from your side. We wanted to follow up on our previous response to your valuable suggestions. Your insights are very important to us, and we are eager to ensure that we have addressed all concerns appropriately.
>
> Warm Regards, :-)

---

### Author Response · Authors · 2024-11-20
**Incorporated feedback and new results**

Thank all reviewers for the thorough review and detailed comments. To address the concerns, we have made the following revisions to the paper:

1. Emphasized the **novelty** of ChartMoE. ChartMoE differs from previous methods in terms of motivation, initialization manner, and MoE application position. We have added this discussion to Appendix `D.1`.
2. Analyzed the **model complexity** and **training complexity** of ChartMoE and included the results in Appendix `D.1`.
3. Added a detailed description of the ChartMoE-Align dataset, including data sources, data processing methods, JSON / Code template, data filtering & cleaning, and data quality control. We have added this discussion to Appendices `C.1-4`.
4. Included the performance results of ChartMoE on other MLLMs (e.g., `LLaVA-7B`) to demonstrate the generalizability of our methods (MoE connector, Diverse Align, etc.). We have added the new experimental results to Appendix `D.2`.
5. Provided a deeper analysis of the experimental results in `Tables 2-3`. By examining fine-grained results, we explain the performance of ChartMoE. We have added the new experimental results and discussions to Appendix `D.3`.
6. Added a discussion on the limitations of ChartMoE, presented in Appendix `D.4`.
7. Improved the ablation experiment settings and added more discussions on the effectiveness of synthetic data based on the reviewers' suggestions. Please refer to the specific discussions with the reviewers.

---

### Meta-Review · Area_Chair_Z7ob · 2024-12-19

**Metareview:**

The paper introduces ChartMoE, a model that enhances complex chart understanding through a Mixture of Expert (MoE) architecture, replacing traditional linear projectors. It also presents the ChartMoE-Align dataset, designed for three specific alignment tasks, demonstrating the model's practical utility. The experimental results are robust and convincing. I recommend that the authors strengthen the presentation of their novel approach in the Introduction to highlight the advancements over existing methods better.

**Additional Comments On Reviewer Discussion:**

Discussion summary during the rebuttal period:

1. Novelty of ChartMoE: Although the Appendix provides further clarification on the novelty of ChartMoE, it still fails to demonstrate that the approach is highly novel. It is recommended that the authors enhance the explanation of ChartMoE's novelty in the **Introduction** to strengthen its motivation.

2. Generalization and contribution: The authors have provided additional experiments and further elaborations, which help in understanding the general applicability and contributions of ChartMoE.

3. Complexity analysis: It has been demonstrated that ChartMoE does not significantly increase computational costs.

2. Data construction details and experiment analysis: More details on data construction and deeper analysis of experiments are discussed, providing greater insight into the methodologies and findings.

---

### Decision · Program_Chairs · 2025-01-22

Accept (Oral)